# Chalcogenides in Perovskite Solar Cells with a Carbon Electrode: State of the Art and Future Prospects

**DOI:** 10.3390/nano14221783

**Published:** 2024-11-06

**Authors:** Maria Bidikoudi, Elias Stathatos

**Affiliations:** Nanotechnology and Advanced Materials Laboratory, Department of Electrical and Computer Engineering, School of Engineering, University of the Peloponnese, 26334 Patras, Greece

**Keywords:** perovskite, solar cells, carbon electrode, chalcogenides

## Abstract

Perovskite solar cells (PSCs) have been on the forefront of advanced research for over a decade, achieving constantly increasing power conversion efficiencies (PCEs), while their route towards commercialization is currently under intensive progress. Towards this target, there has been a turn to PSCs that employ a carbon electrode (C-PSCs) for the elimination of metal back contacts, which increase the cost of corresponding devices while at the same time have a severe impact on their stability. Chalcogenides are chemical compounds that contain at least one chalcogen element, typically sulfur (S), selenium (Se), or tellurium (Te), combined with one metallic element. They possess semiconducting properties and have been proven to have beneficial effects when incorporated in a variety of solar cell types, including dye sensitized solar cells (DSSCs), quantum dot sensitized solar cells (QDSSCs), and Organic Solar Cells (OSCs), either as interlayers or added in the active layers. Currently, an increasing number of studies have highlighted their potential for achieving high-performing and stable PSCs. In this review, the most promising results of the latest studies regarding the implementation of chalcogenides in PSCs with a carbon electrode are presented and discussed, merging two research trends that are currently on the spotlight of solar cell technology.

## 1. Introduction

Perovskite solar cells (PSCs) have been considered the next big thing in solar cell technology for the past 10 years, and significant amount of research and financial resources have been devoted to their advancement, which has led to the successful transfer of this technology from lab-scale devices to perovskite solar modules (PSMs) with power conversion efficiencies (PCEs) that exceed 20% [1].

In order to overcome significant drawbacks that hinder the commercialization of PSCs, which mainly concern the use of noble metals as the counter electrode, a new type of PSCs has been developed and studied intensively in recent years, where the noble metal electrode is substituted by a carbonaceous electrode. This electrode is typically prepared after the deposition and annealing of a carbonaceous starting paste, which is low-cost and easy to prepare, while it can be deposited with a wide variety of industrially compatible methods, such as screen printing and blade coating. The resulting electrode is distinguished by a high chemical stability over the reactivity of halogens present in the perovskite, which is one cause of degradation in PSCs with a metal electrode, and a cost of fabrication which is 80% lower than that of the respective metal electrode, leading to an overall significant reduction in the resulting devices’ cost of fabrication [2]. This type of PSCs is referred to as C-based or C electrode perovskite solar cells (C-PSCs).

Chalcogenide materials have a broad range of chemical compositions, where any alteration leads to significant changes in their physical and chemical properties, including their energy bandgap, carrier mobility, absorption coefficients, and chemical stability, which are some key factors to be taken into consideration in the solar cell technology. This tunability of chalcogenides, along with their potential low-toxicity and availability in the earth’s crust, has made them ideal candidates for application in photovoltaics. Starting from thin-film solar cells, where chalcogenides have emerged as a highly efficient active material-absorber [3], and moving to the incorporation of chalcogenides in a variety of solar cell types as different components according to their properties, where they have also exhibited promising results, it is now the time where the potential of chalcogenides is explored, as per their ability to achieve PSCs with advanced properties [4,5,6].

This review intends to present the latest advancements in the incorporation of chalcogenides in PSCs with a C electrode. The motivation has been on the one hand the highly promising results that have been obtained by the implementation of chalcogenides in PSCs with a metal electrode, which achieves high performance and stability, and on the other hand, the noticed increasing number of reports and increasing funding on the research on chalcogenides in PSCs, which is currently moving to C-PSCs as well. The focus on PSCs with a C electrode, a PSC structure that is gaining an increasing amount of attention, owing to the favorable properties that make it the most suitable and viable choice for the future commercialization of large-area PSCs, is what distinguishes this review from the related reviews that already exist in the literature. The lack of any relevant reports so far is a gap in the literature that is intended to be bridged by this work.

In the first part of the review, a brief mention is made of the PSC technology and the description of the C-PSC structure, singularity, and advantages. Then, a definition of chalcogenides is presented, along with their properties and classification, followed by a description of their background use in solar cells in general. Then, the main part focuses on the implementation of chalcogenides in PSCs with a C electrode, which is divided into four categories based on the chalcogen in the chalcogenide structure: 1. sulfides, 2. selenides, 3. ternary, and 4. quaternary. Finally, the prospects and challenges of chalcogenide incorporation in C-PSCs are discussed, and suggestions about future directions are made. We believe that this review will be of high added value to the constantly evolving field of carbon electrode perovskite solar cells, since it deals with one of the most up-to-date developments in the field and can become the motivation for further exploration and improvement of C-PSCs, which are the most prominent PSC structure for the future commercialization of this highly valued technology.

## 2. Perovskite Solar Cells (PSCs)

In 2009, a new type of material was used as a novel, hybrid organic–inorganic absorber in dye sensitized solar cells (DSSCs), namely lead halide perovskite compounds methylammonium lead bromide (CH_3_NH_3_PbBr_3_) and methylammonium lead iodide (CH_3_NH_3_PbI_3_) [7]. The motivation for the incorporation of perovskite compounds as a replacement of the typically used organic dye molecules that have used so far as sensitizers has been their intriguing optical and excitonic properties, combined with the abundance of the elements in their composition. The perovskite sensitized solar cells employing these two compounds have yielded power conversion efficiencies (PCEs) as high as 3.81%, which is lower compared to the corresponding PCE obtained with organic dyes. However, the properties of perovskite absorbers have triggered the research interests into further investigations. Three years later, the CH_3_NH_3_PbI_3_ lead halide perovskite was incorporated as the sensitizer in solid-state DSCCs, employing a hole transport material, namely 2,2′,7,7′-tetrakis-(*N*,*N*-di-*p-*methoxyphenyl-amine)-9,9′-spirobifluorene (spiro-MeOTAD), to replace the liquid electrolyte and eliminate the problems that it was imposing on the stability of the device, which achieved a PCE of 9.7%, highlighting the great potential of the new material as a light absorber in solid-state solar cells. These solid-state solar cells have constituted the archetype of this new class of thin film solar cells, perovskite solar cells (PSCs), which, after extensive and intensive research over the years, have now managed to be the highlight of solar cells, exhibiting PCEs that exceed 25%, enabling the fabrication of large-area solar panels with PCEs that are comparable to silicon solar cells (Si), whereas a series of products have also been launched, introducing this technology to the market [1,8].

The typical structure of PSCs includes an electron transport layer (ETL) and a hole transport layer (HTL), which are deposited on a conductive substrate, such as fluorine doped tin oxide (FTO) glass. The perovskite absorber is implemented between these two charge transfer materials and the circuit closes with a counter electrode, which is typically gold (Au) or silver (Ag). The main drawbacks that PSCs suffer from, which inhibit their large area and wide application, are as follows: i. their thermal and ambient stability under storage and illumination; ii. their high cost of materials and fabrication methods, and iii. the toxicity of elements of the highest-performing PSCs, which are Pb-based [9,10]. Among these, both the stability and the high cost are directly related to the metal electrode employed, which on the one hand is expensive and is deposited by costly methods that are not appropriate for large-area applications, and on the other hand, they are not chemically stable over the halide elements present in the perovskite compound and are gradually corroded. An additional factor contributing to these limitations that PSCs face is about the hole transport materials that are typically used, which are mainly organic or polymeric and suffer from limited stability under irradiation, while they are also of high costs.

Both of these factors that limit the wide application of PSCs and their turning into a viable innovation for the future of solar technology have been successfully mitigated by the introduction of perovskite solar cells with a carbon electrode, namely carbon electrode perovskite solar cells (C-PSCs). In this device configuration, the metal electrode is replaced by a C electrode, prepared by starting pastes or slurries, comprising carbon materials combined with binders and organic solvents, which are deposited using simple and low-cost methods and create the C electrode after annealing [11,12] (Figure 1a). Carbon electrodes have proven to be very effective as counter electrodes for perovskite solar cells, and the rapid increase in power conversion efficiency achieved, from 6% to 22% in just 10 years (Figure 1b) [13], has now led to the creation of a dedicated industry that deals with research on these cells, which have a range of advantages compared to the typical metal electrode-based structure.

The two main configurations of C-PSCs are distinguished by the annealing temperature that is applied for the preparation of the C electrode, and they include the low-temperature C-PSCs (LT-CPSCs) and high-temperature C-PSCs (HT-CPSCs), respectively. In LT-CPSCs, a C paste is deposited on the HTL by painting, blade coating, or printing methods, and the electrode is formed after annealing at temperatures below 100 degrees. On the other hand, in HT-CPSCs, a triple mesoscopic stack is formed by a mesoporous ETL, typically TiO_2_; a mesoporous insulating layer that prevents the contact of the resulting counter electrode with the ETL, typically ZrO_2_; and a porous, thick C electrode that is deposited on top of the TiO_2_/ZrO_2_ bilayer, mostly by blade coating, in the form of C paste, which is further treated at high temperatures (350–400 degrees). The perovskite is then inserted in C-PSCs by infiltration of a small amount of perovskite precursor solution through the triple mesoscopic stack, and the perovskite crystals are formed through subsequent annealing. Despite the major differences in the device configuration, the two structures have some common features, which are of great importance for the transfer of this technology to large areas and the commercialization of PSCs of this type: the deposition methods are simple, low-cost, and industrially compatible, while they also achieve a minimum waste of materials, hence a low cost of fabrication. Both configurations have also proven to be efficient after the elimination of the HTL, which has a great effect on both the cost and the stability of the devices, while they can be prepared entirely under ambient conditions, with no control facilities (e.g., glove box, clean room) required. Moreover, the chemical inertness of carbon, which prevents the degradation of the electrode overtime, combined with the hydrophobic nature of carbon, which prevents the degradation of the perovskite layer, gives a significant advantage of C-PSCs over metal electrode PSCs, since they have proven to be of high thermal, environmental, and operational stability [14,15,16,17]. Adding to the above, because of the considerably lower cost of C electrodes compared to metal electrodes, C-PSCs are the most suitable candidates for commercially available perovskite solar modules (PSMs), which is also proven by the increasing number of PSM fabrication companies that switch to C electrodes in order to achieve a viable product.

## 3. Chalcogenides

### 3.1. Definition and Properties

The class of materials that are defined as chalcogenides include the chemical compounds that consist of at least one anion from the group 16 elements of the periodic table, namely the chalcogens, most commonly sulfur (S), selenium (Se), and tellurium (Te), combined with an ion of a more electropositive element. These ions form covalent bonds, and the resulting materials, either amorphous or crystalline, are semiconductors with energy band gaps ranging from 1 to 3 eV, which can be tuned according to their composition.

The classification of chalcogenides has been proposed in three different directions as shown in Figure 2 [18]:According to the type of cation in their structure, they are classified in three main categories: alkali or alkaline earth, transition metal, and main-group chalcogenides.According to the number of their components, they are classified as binary, ternary, and quaternary structures.According to the number of chalcogen ions, they are classified as monochalcogenide, dichalcogenide, and trichalcogenide.

The ability to tune their energy bandgap, along with their excellent optical properties, such as a high refractive index and photosensitivity, and notable electrical conductivity has led to the extended study of chalcogenides as potential candidates for a variety of applications, including photonic devices, thermoelectric devices, thin-film electronics, catalysis [19,20,21,22], superconductivity, and solar cells [23].

It is worth noting that a separate class of chalcogenide materials, namely chalcogenide perovskites, are currently receiving an increasing amount of attention as a new class of materials for energy-related applications [6,24,25,26]. These chalcogenides have the typical structure of perovskite compounds, which is ABX_3_, where A is typically a group II cation, such as Barium (Ba) and Calcium (Ca); B is a group IV transition metal, such as Zirconium (Zr); and X is a chalcogen, and they are considered the most promising all-inorganic, alternative to hybrid organic–inorganic Pb-based perovskites that have been on the forefront of research for the past 10 years. This is attributed to their favorable optoelectronic properties combined with lower toxicity and higher thermal stability compared to their Pb-based counterparts.

Table 1 summarizes the optical and electronic properties of some of the most widely investigated chalcogenides that have been used in solar cells so far.

### 3.2. Implementation in Solar Cells

Metal chalcogenides, including metal, transition metal, semi-metal, and rare-earth metal chalcogenides, irrespective of the number of constituents and chalcogens, possess electronic structures that can be tuned through a series of methods, some of which are thickness modification; defect engineering; intercalation of atoms, molecules, and ions in their lattice; and chemical composition modifications. All of the aforementioned alterations have an impact on their optoelectronic properties, which consequently affect and determine their performance once implemented in solar cells. More specifically, the quantum confinement effect that appears when moving from bulk chalcogenides to low-dimensional chalcogenide structures (e.g., 2D) is responsible for the following: i. the transition of their bandgap from indirect to direct, with a simultaneous enlargement; and ii. the appearance of strong excitonic effects, attributed to the stronger electrostatic force interactions at lower thicknesses. On the other hand, defect engineering can lead to higher photoluminescence intensities, while the intercalation of various species in the lattice of chalcogenides can induce the transition from semiconducting to conducting and enhance optical transmission. Finally, by alloying metallic and semiconducting chalcogenides and fine tuning their chemical composition, a fine tuning of the resulting alloy’s bandgap can be achieved [43,44,45,46,47].

Owing to their versatile properties, chalcogenides have been used in all components of solar cells: as electron transport layers, as hole transport layers, as the absorber-active material, and as the counter electrode in next-generation solar cells. A variation of the metallic element in the chalcogenide, typically combined with sulfur (S) and selenium (Se), results in variations in the optoelectronic properties of the resulting chalcogenide, with prominent effects on the energy levels and the type of mobility of the resulting materials and corresponding thin films [43,48,49]. The performance of chalcogenides’ application in solar cells exhibits the following trends:Transition metal chalcogenides (TMDs), such as molybdenum disulfide (MoS_2_), molybdenum diselenide (MoSe_2_), and tungsten diselenide (WSe_2_), have a superior hole mobility while they can also produce homogeneous films with tunable properties [50]. They have been, therefore, mainly used as hole transport materials (HTMs) to prepare the hole transport layers (HTLs) for organic photovoltaics (OPVs) and perovskite solar cells (PSCs).Metal chalcogenides, such as lead (Pb)-based and copper (Cu)-based ternary and quaternary, together with transition metal cadmium (Cd)-based and the semi-metal antimony (Sb)-based chalcogenides, have exhibited excellent performance as absorbers for thin-film solar cells and quantum dot solar cells (QDSSCs).Transition metal cadmium (Cd)-based sulfides and selenides have exhibited satisfactory performance as electron transport layers (ETLs) applied in perovskite solar cells (PSCs), organic photovolatics (OPVs), and antimony chalcogenide solar cells (Sb-CSCs).Transition metal cobalt (Co) and nickel (Ni) sulfides, along with metal copper (Cu) sulfide chalcogenides, have achieved high-performance photoelectrochemical solar cells, which include dye sensitized solar cells (DSSCs) and quantum dot sensitized solar cells (QDSSCs), when applied as counter electrodes.

#### 3.2.1. Chalcogenides as Hole Transport Materials (HTMs)

Chalcogenides are well known to have tunable properties, including their bandgap values, but also depending on their structure, they are distinguished by p-type conductivity, which makes them excellent hole transport material candidates. Combined with enhanced stability, as applies for all inorganic molecules, compared to the organic HTMs that have been extensively used so far, chalcogenides have exhibited highly promising results. The most widely applied and efficient chalcogenides that have been applied as HTMs have been transition metal dichalcogenides (TMDs), including molybdenum disulfide (MoS_2_), tungsten diselenide (WSe_2_), and tungsten disulfide (WS_2_) [51], as well as copper (Cu)-based compounds, including copper iodide (CuI),copper sulfide (CuS), and the ternary and quaternary copper-based derivatives [52]. Chalcogenides have typically been used as the hole transport layer in thin-film solar cells, organic photovoltaics (OPVs), and perovskite solar cells (PSCs).

In OPVs, the chalcogenides that have been widely used as HTLs are mainly transition metal chalcogenides, such as MoS_2_, WSe_2_, and WS_2_, which have achieved optimum PCE values that exceed 20% [30,53,54,55,56,57]. Copper compounds have also been tested, namely CuI and CuS; however, the corresponding device PCEs remain low, while the complex and costly deposition methods of Cu compounds, compared to TMDs, which are solution processable, make them less attractive for extensive applications and investigations [58,59].

In PSCs, the most promising results have been obtained with copper derivatives, while TMDs have seen a significant rise in the investigation of their properties and potential application as alternative, inorganic HTMs, which has led to a significant increase in the PCE values obtained, exceeding 20% [60].

In photoelectrochemical solar cells, binary sulfides, mainly sodium sulfide (Na_2_S), have been typically employed to prepare the posysulfide electrolyte used in quantum dot sensitized solar cells (QDSSCs).

The implementation of chalcogenides as inorganic hole conductors in solar cells is very wide and multidirectional and exceeds the scope of this review. However, Table 2 summarizes some indicative works that have been reported and achieved significant results. The application of chalcogenides in carbon electrode-based perovskite solar cells is discussed in detail further along the manuscript.

#### 3.2.2. Chalcogenides as Light Absorbers

The favorable tunable, direct bandgap of chalcogenides, ranging from 1 to 2 eV, and the high absorption coefficient of over 10^4^ cm^−1^, combined with high thermal stability and low toxicity [25], make chalcogenides highly promising materials to be employed as absorbers. And in fact, in the last 10 years, a significant amount of research has been devoted to the investigation of their properties and potential as light harvesters, leading to notable improvements in their power conversion efficiency values, as shown by the PCE evolution chart in Figure 3 [74].

Chalcogenides have been widely and very effectively used as absorbers in a distinctive class of thin-film solar cells, namely chalcogenide solar cells, as well as in quantum dot sensitized solar cells (QDSSCs). In thin-film solar cells, the typical structures that have been employed and yielded the highest power conversion efficiencies (PCEs) are the binary Cadmium Telluride (CdTe), Cadmium Selenide (CdSe), and lead sulfide (PbS) compositions as well as the more environmentally friendly, Cd-free, and Pb-free quaternary copper-based structures, such as Copper Indium Gallium Sulfide (CIGS), Copper–Zinc–Tin Sulfide (CZTS), Copper–Zinc–Tin Selenide (CZTSe), and Copper Zinc Tin Sulfide Selenide (CZTSSe). The highest PCEs that exceed 20% have been obtained so far in CdTe solar cells, owing to the bandgap of CdTe, which makes it a very promising absorber material (~1.44 eV) [75] (Figure 4a). However, the toxicity issues, along with the high fabrication cost, make the wide use of CdTe questionable. The copper-based chalcogenides, in particular CIGS, seem to be a highly promising alternative to Cd solar cells, since the CIGS solar cells have been able to deliver PCEs of over 23%, with no toxic elements in their structure, but with the drawback of complexity of preparation and scarcity of elements, which could hinder large-scale production [76] (Figure 4b). CIGS has also been implemented in tandem structure solar cells that have exhibited PCEs of over 29% [77].

Recently, a new class of chalcogenide solar cells has received an increased level of attention due to their reduced toxicity, and these are antimony-based (Sb) chalcogenide and antimony (Sb) perovskite solar cells. In this type of devices, Sb-based compounds act as the absorber, while the configuration of the solar cells is versatile and can have a mesoporous structure, resembling sensitized solar cells (DSSCs and QDSSCs), the planar superstrate structure, or the planar heterojunction structure (Figure 5a). Sb chalcogenide solar cells have reached PCEs exceeding 10%, while a lot of effort has been made in order to further increase these values (Figure 5b) [80,81].

On the other hand, chalcogenides have also been widely used in quantum dot sensitized solar cells (QDSSCs) as absorbers. These types of solar cells have the advantages of a high absorption coefficient, fast charge separation, photo durability, and stability, compared to the dyes typically used in dye sensitized solar cells (DSCCs) as sensitizers, with multiple exciton generation capability and bandgap tunability through compositional engineering of the chalcogenide absorber. The most efficient chalcogenide structures have been metal chalcogenides, which can be synthesized and deposited by simple, solution-based methods, such as the hot-injection, chemical bath deposition (CBD), electrophoresis, and successive ionic ligand exchange and reaction (SILAR), which yield colloidal, narrow-sized, highly monodispersed quantum dots with a wide range of emissions (Figure 6a,b) [82].

The binary QDs that have most widely been used are lead sulfide (PbS) Cd-based, mainly CdS, CdSe, and CdTe, while the highest PCEs, reaching 10%, have been demonstrated with cadmium-based alloyed QDs, such as CdS_x_Se_1−x_ and CdSe_x_Te_1−x_. The most efficient devices that have been exhibited so far employ quinary alloyed Zn-Cu-In-S-Se QDs, with PCE exceeding 13% (Figure 6c) [83], while a variety of complex ternary, quaternary, and quinary structures that are Pb-free and Cd-free have also been reported. However, their PCEs are limited to <10% [84]. Despite their favorable and theoretically promising properties, QDSSCs suffer from severe recombination phenomena, stemming from energy band misalignment, which restrict their efficacy, while practical limitations, which include the toxic elements that they comprise and the instability of the electrolyte-counter electrode couple that can be employed, make QDSSCs far from a viable technology, especially for large-area and large-scale applications.

Finally, transition metal chalcogenides, such as WSe_2_, MoS_2_, and WS_2_, have been implemented as additives in the active layer of OPVs, achieving slight improvements in the obtained PCEs. However, the results are not as promising as for other 2D materials, mainly graphene-related materials (GRMs) [51], whereas the PCEs achieved remain lower than 10%.

#### 3.2.3. Chalcogenides as Electron Transport Materials (ETMs)

Even though reports of chalcogenides being used as electron transport materials (ETMs) for the fabrication of solar cell electron transport layers (ETLs) are far less than their successful application as HTLs and absorbers, there are still certain types of solar cells, where chalcogenides with high electron mobility have proven to be a promising alternative to the typically used ETL materials, such as wide bandgap semiconducting oxides (e.g., TiO_2_, SnO_2_, ZnO) and fullerene derivatives (e.g., Phenyl-C61-butyric acid methyl ester-PCBM). These are mainly perovskite solar cells, both n-i-p structured and p-i-n structured, where metal and transition metal chalcogenides have been used to prepare the ETL and the corresponding devices have yielded PCEs ranging from 13 to >20% [85,86]. Chalcogenides have also been employed as ETLs in organic photovoltaics, with selenium-based compounds being the most efficient, achieving PCEs that exceed 9% [53,54]. Some reports also refer to the effectiveness of CdS as the ETL in antimony-based chalcogenide solar cells (Sb-CSCs), while QDSSCs with PbS as the ETL have also achieved PCEs as high as 10.5% [87]. Overall, the electron-conducting properties of chalcogenides have not yet been extensively explored, given that there exist materials with higher electron mobility than chalcogenides, which are less toxic, cheap, and abundant and can be applied with various simple and low-cost methods, while they are able to produce solar cells with high PCEs. Figure 7 summarizes some representative applications of chalcogenides as ETLs that have been reported in the literature. However, it should be highlighted that a deeper investigation of chalcogenides as ETLs could include more data, which is out of the scope of this work.

#### 3.2.4. Chalcogenides as Counter Electrodes

Metal and transition metal chalcogenides are low-cost, abundant materials with high catalytic activity and conductivity, which can be synthesized with a variety of simple and scalable methods and can be tuned as per their properties by simple alterations in their chemical composition. This makes them good candidates for counter electrodes in photoelectrochemical solar cells that are composed of a semiconductor working electrode, an electrolyte bearing a redox couple, and a counter electrode. These include dye sensitized and quantum dot sensitized. Chalcogenides have been successfully incorporated in both types of solar cells as the counter electrode to replace the typical platinum counter electrode used in dye sensitized solar cells (DSSCs), which is costly and scarce, and the copper brass counter electrode typically used in quantum dot sensitized solar cells (QDSSCs), which lacks stability. In DSSCs, the PCEs that have been obtained are close to the PCEs obtained with the Pt electrode, which demonstrates the capability of chalcogenides to be used as a cheap alternative to Pt electrodes for photoelectrochemical cells. The most promising results have been obtained with cobalt binary metal chalcogenides, such as CoSe, and with ternary chalcogenides, such as Ni-Co-Se and Ni-Co-S, where PCEs exceeding 10% have been recorded [88], while binary transition metal chalcogenides, such as NiS, have achieved PCEs that exceed the PCE obtained with Pt electrodes. Moreover, they have shown promising results as flexible and transparent electrodes, which broadens their potential applications [89,90]. In QDSSCs, the most promising structures have been transition metal sulfides, which have high catalytic activity, low charge transfer resistance and a good affinity with the typically used polysulfide electrolute, and high operational stability, at a much lower cost [88,91].

## 4. Incorporation in Carbon Electrode PSCs (C-PSCs)

### 4.1. Sulfides

#### 4.1.1. Molybdenum Disulfide (MoS_2_)

Molybdenum disulfide (MoS_2_) is probably the most widely studied among the family of sulfides. Its favorable properties as a surface modifier and the catalytic activity has been originally reported in DSSCs, while it is also commonly employed in metal electrode PSCs, mainly of the inverted structure. In C-PSCs, the effect of MoS_2_ has been investigated in a variety of applications: as a buffer layer over the perovskite, to passivate the perovskite grain boundaries, as an additive to improve the electron transport properties of the TiO_2_ electron transport layer (ETL), as an additive in the bulk perovskite, as a novel composite counter electrode, and as the hole transport layer in Pb-free perovskites. In all applications, the addition of MoS_2_ has yielded improved photovoltaic parameters, which highlights the versatility of this material and its great prospects for further investigation.

Duan et. al. first reported on the use of molybdenum disulfide (MoS_2_) in C electrode-based inorganic perovskite solar cells. In particular, they used a simple reflux method to produce MoS_2_ quantum dots (QDs), which were then incorporated as a buffer layer at the CsPbBr_3_ perovskite/carbon interface [92]. The few-layered MoS_2_ QD film proved to promote the film formation capability and to increase the conductivity of the resulting film. Moreover, the MoS_2_ QDs had a positive effect on the recombination dynamics. By passivating the grain boundaries and by modifying the energy level distribution, the MoS_2_ QD devices exhibited an increased PCE of 6.8% under 1 sun illumination, compared to 5.68% for the reference device, with no buffer layer. This increase was mainly attributed to an increase in both the Jsc (from 5.73 to 6.55 mA/cm^2^) and the Voc (from 1.285 to 1.307 V), indicating the accelerated charge extraction, with simultaneous hindering of electron-hole recombination at a CsPbBr_3_/carbon interface for the MoS_2_ QD-treated solar cells. These results were also supported by steady-state photoluminescence (PL) measurements, where both quenched PL intensity and a reduced calculated carrier lifetime confirmed the promoted hole extraction from the perovskite to the C electrode and the passivation effect at the perovskite/C interface after the insertion of a MoS_2_ QD layer (Figure 8).

Zhou et al. incorporated a series of transition metal dichalcogenide quantum dots (TMDCs QDs), including MoS_2_ QDs, in low-temperature processed TiO_2_ ETLs (L-TiO_2_) and achieved enhanced electron transport properties upon illumination [93]. After the addition of the QDs in the ETL, the conductivities of the L-TiO_2_:MoS_2_ films were enhanced, compared to the pristine L-TiO_2_, which was directly related to the increase in the electron mobility of L-TiO_2_:MoS_2_ from 5.35 × 10^−4^ to 7.30 × 10^−4^ cm^2^ V^−1^ s^−1^ and a simultaneous decrease in the electron trap-state densities of L-TiO_2_:MoS_2_ from 7.27 × 10^17^ to 3.04 × 10^17^ cm^−3^ under illumination, contrary to the value of pristine L-TiO_2_, which showed a negligible decrease. The modified ETLs were paired with the inorganic CsPbBr_3_ perovskite in HTL-free C-PSCs employing a low-temperature C electrode. The PCE that has been obtained has been as high as 9.22% for the L-TiO_2_:MoS_2_ devices, which is significantly higher than the 8.25 of the reference C-PSCs. The enhanced performance has been a result of increased short circuit current density, originating from the improved charge transport and the simultaneous increase in the open circuit voltage originating from reduced recombination. Finally, the C-PSCs incorporating the transition metal disulfide QDs exhibited good stability, retaining over 90% of the initial Voc, Jsc, PCE, and FF values over 10 days of storage in air.

Han et al. also used the inorganic CsPbBr_3_ perovskite in low-temperature C electrodes, HTL-free PSCs, employing the low-temperature SnO_2_ ETL, where MoS_2_ were added in the perovskite precursor bulk as an additive [94]. A series of concentrations have been explored, which have all resulted in the formation of CsPbBr_3_–MoS_2_ hybrid films with stronger light absorbance than the pristine CsPbBr_3_ film. The hybridization of CsPbBr_3_ with MoS_2_ has also led to reduced trap density and an acceleration of the hole extraction process. Finally, the addition of MoS_2_ has improved the crystalline quality of the perovskite films and has led to inhibition of the formation of the impurity phase of cesium lead bromide. By improving all the electrical parameters of the resulting C-PSCs, the concentration of 0.4 mg/mL MoS_2_ exhibited the highest PCE of 7.87%, compared to 6.62% of the pristine CsPbBr_3_ perovskite. In addition, the C-PSCs with the hybrid absorber preserved 98% of their initial PCE value under outdoor ambient air for 30 days (Figure 9).

In a different approach, Wei e.al. constructed a compound of an MoS_2_ and MoP in situ composite (N, P co-doped carbon nanospheres MoS_2_–MoP/NPC) as an alternative, hybrid carbon electrode, which was further evaluated as per the performance as a counter electrode for HTL-free C-PSCs, employing the inorganic CsPbBr_3_ perovskite [95]. The introduction of MoS_2_ enhanced the p-type properties of the composite carbon electrode, therefore increasing the hole mobility; at the same time, the conductivity was also increased, and the work function was down-shifted. Moreover, the N, P, and S components of the MoS_2_–MoP/NPC nanospheres exhibited a passivation effect on the defects of the perovskite film, resulting in reduced non-radiative recombination. All the above combined led to C-PSCs employing the composite electrode to exhibit PCEs as high as 10.13% and a notable open-circuit voltage of 1.638 V, which is significantly higher than the nanocarbon electrode-based devices, delivering a 6.93% PCE and an open-circuit voltage of 1.476 V (Figure 10).

Driven by the promising results that MoS_2_ has exhibited in typical hybrid organic inorganic perovskite (HOIP)-based C-PSCs and taking advantage of the high charge carrier mobilities (e.g., 470 cm^2^ V^−1^ s^−1^ of electrons, 480 cm^2^ V^−1^ s^−1^ of holes) and great chemical stability of MoS_2_ nanoflakes, Pang et al. [96] reported on the successful fabrication of all-inorganic Pb-free PSCs, with a C electrode and the double Cs_2_AgBiBr_6_ perovskite, where MoS_2_ nanoflakes were used as the hole transport layer. The Pb-free PSCs with MoS_2_ nanoflakes as an HTL exhibited significantly higher PCE values compared to the corresponding devices using the typical polymer of triarylamine PTAA (Poly[bis(4-phenyl) (2,4,6-trimethylphenyl)-amine]) as a hole transport layer, owing to the great enhancement of the obtained short circuit current density. This is attributed to the ability of MoS_2_ nanoflakes to increase the optical absorption, as confirmed by the optical absorbance spectra obtained for the reference, PTAA, and MoS_2_-based PSCs. Moreover, the crystallinity of the devices with MoS_2_ was proven to increase, compared to the reference, while the carrier mobility was shown to improve, owing to the MoS_2_ nanoflakes being embedded at the grain boundaries of Cs_2_AgBiBr_6_, leading to the enhanced performance of the Pb-free PSCs. Finally, the MoS_2_ nanoflakes had a positive impact on the stability of the devices, and the PCE of MoS_2_-based device decreased to 92% after 23 days of exposure to ambient conditions, while the corresponding devices with PTAA and pristine showed a decrease to 85% and 72%, respectively (Figure 11).

#### 4.1.2. Tungsten Disulfide (WS_2_)

Tungsten disulfide (WS_2_) is a transition metal dichalcogenide (TMD) with similar properties as MoS_2_, including high thermal and chemical stability. In PSCs, it has most commonly been employed in inverted structure devices to serve as a type of cascade promoting the epitaxial growth of perovskite films on the hole transport layer (HTL), which reduces the defect formation, thus reducing the interfacial recombination and enhancing the charge extraction process. In C-PSCs, the use of WS_2_ as an interlayer to achieve the epitaxial growth has also been reported. However, its potential use has not been restricted at this application, and WS_2_ has also been presented as a novel hole transport material and as an additive in the antisolvent treatment during the formation of the perovskite films.

In particular, taking advantage of WS_2_ films’ ability to promote the Van der Walls preferential growth of hybrid organic inorganic perovskites (HOIPs), which has led to noteworthy results in PSCs employing a metal electrode, Zhou et al. [97] prepared a WS_2_/perovskite heterostructure to address the problem of residual strain in inorganic perovskite films, which compromises carrier dynamics and device stability. The authors introduced 2D WS_2_ nanoflakes into the ETL/perovskite interface, where the ETL was a composite of SnO_2_-TiO_x_Cl_4−2x_ and the perovskite was the inorganic CsPbBr_3_ (Figure 12a). After the introduction of WS_2_ nanoflakes, a self-assembled WS_2_/CsPbBr_3_ heterostructure was created, which was shown to act as a type of template of van der Waals epitaxy, producing high-quality CsPbBr_3_ perovskite films. The perovskite films that were grown on the WS_2_ interface exhibited enlarged grain sizes and fewer grain boundaries compared to the films grown on the bare ETL, with a preferential orientation perpendicular to the substrate, as shown in Figure 12b. This heterostructure promoted the interfacial tensile strain release and led to an increase in the PCE of PSCs, 9.27% to 10.65%, which was attributed to the significant increase in the Jsc and Voc values upon interfacial regulation. The devices exhibited a remarkable Voc value of 1.7 V and long-term stability of 10,000 s of light soaking, retaining 90% of their initial PCE, and over 120 days of storage in ambient conditions, retaining over 80% of their initial PCE (Figure 12c,d).

In 2023, Sui et al [98] demonstrated the efficient hole transport properties of WS_2_ by preparing a composite of WS_2_/AgIn_5_S_8_ quantum dots (QDs), which was used as a novel hole transport material (Figure 13). The PSCs fabricated with the use of the polysulfide QDs composite as the HTM and the inorganic CsPbBr_3_ perovskite exhibited PCEs of up to 10.24%, as opposed to the control device, which exhibited a PCE of 6.68% and a notable open-circuit voltage (Voc) of 1.627 V, which arose from the enhanced advanced charge transfer and decreased charge recombination. The beneficial effect of the polysulfide composite was described to be three-fold: i. upon incorporation of AgIn_5_S_8_ QDs in WS_2_ QDs, the sulfur vacancies of the latter are passivated and a Type-II band is formed, which leads to a higher charge transfer capability; ii. the non-coordinated Br ion defects of the perovskite film are reduced by the formation of Ag-Br and In-Br bonds, stemming from the Ag and In elements; and iii. the surface Pb ions of the perovskite film are passivated, while at the same time interact with the S elements to create a Pb-S-W track, which favors hole extraction. Finally, the formation of Pb-S, Ag-Br, and In-Br bonds in the WS_2_/AgIn_5_S_8_ QDs composite increased the hydrophobicity, contributing to the increase in the PSCs stability in ambient conditions (85% RH at 85 °C), where a 93% maintenance of the PCE after 720 h of exposure was recorded.

Finally, WS_2_ quantum dots have been used in PSCs with a C electrode, employing the organic–inorganic methylammonium lead triiodide (MAPbI_3_) perovskite, as an additive in the antisolvent treatment stage [99]. The novelty of this work lies in the fabrication method of the WSQDs, which were generated using pulsed laser irradiation directly in the antisolvent ethyl acetate. The general benefits of this method are that it is easy and quick; it can produce nanoparticles in a few minutes, the sizes of which can be modified by simply changing the laser fluence; and the obtained material is of high purity and ligand-free. The specific beneficial feature, for the application in PSCs, is that the desired nanoparticles are able to be formed directly in the antisolvent and used without any further procedures, leading to high-quality, WSQDs-incorporated perovskite films. The introduction of WS_2_ quantum dots with this method has led to devices with enhanced overall performance, from 12.15% for the pristine device to 15.94% for the device with the optimum concentration of WSQDs of 0.1 mg/mL. A particular increase in the Voc and FF values suggests reduced recombination and a fast charge transport, attributed to reduced defects and superior band alignment for the devices prepared with the WSQDs-bearing antisolvent.

#### 4.1.3. Cadmium Sulfide (CdS)

Cadmium sulfide, which is an n-type semiconductor, with a favorable 2.3 eV direct band gap, has long been used in the form of quantum dots as an inorganic semiconductor sensitizer of TiO_2_ in DSSCs, with the exciting feature of multiple exciton generation upon illumination, which could result in high values of photocurrent obtained from the devices. However, the PCE in CdS QD sensitized solar cells has remained low, despite the yearly efforts towards their improvement, mainly owing to the low fill factor and Voc values obtained.

Liu et al. took advantage of CdS’s favorable properties, including a high electron mobility (4.66 cm^2^ V^−1^ s^−1^) and similar conduction band position to TiO_2_, and they used it as an alternative to the TiO_2_ electron transport layer for C-PSCs, employing a Cs/MA/FA mixed cation, an I/Br mixed halide perovskite [100]. The CdS layer was deposited by thermal evaporation in order to avoid the hazardous wet chemistry techniques, considering the toxicity of Cd and the optimized devices, with a CdS ETL of 100 nm exhibiting PCEs as high as 13.22%. In order to improve the device performance, the authors employed an interlayer of the fullerene derivative PCBM (phenyl-C61-butyric acid methyl ester) at the ETL/perovskite interface. After this modification, the PCE reached 14.28%, a value comparable to the corresponding TiO_2_ ETL-based device, which exhibited a PCE of 14.38%. One main advantage of switching to CdS as an ETL, particularly with the fabrication method of thermal evaporation, as suggested in this work, is that the fabrication process can be performed at low temperatures (<100 °C), which allows for the fabrication of flexible C-PSCs as well. The authors fabricated a flexible C-PSC using the configuration of PEN/ITO/CdS/PCBM/Perovskite/CuPc/Carbon and achieved a device performance of a maximum PCE of 9.56%, an FF of 0.66, a *J*_SC_ of 15.44 mA/cm^2^, and a *V*_OC_ of 0.94 V (Figure 14).

In a different approach, Xu et al. [101] prepared all-inorganic C-PSCs, where a heterojunction of PbS/CdS was used as a passivating layer at the perovskite/C interface. This modification of the CsPbI_1.5_Br_1.5_ surface yielded C-PSCs with significantly enhanced electrical parameters, such as a Voc of 1.31 V, an FF of 0.77, a Jsc of 13.47 mA/cm^2^, and an overall PCE as high as 13.65%, whereas the control device PCE was limited to 10.62%. This improvement was explained by the authors as arising from the capability of the PbS/CdS heterojunction to effectively inhibit ion migration and reduce the halide defects and dangling bonds on the surface of the perovskite film, which decreases the trap density and inhibits a recombination of charges. Moreover, the C-PSCs employing the PbS/CdS heterojunction exhibited increased stability, retaining over 90% of their PCE after 1200 h of storage in an ambient environment (RH = 30%) and 87% of the initial PCE in a nitrogen atmosphere at 85 °C for 400 h, which is attributed to the high hydrophobicity of sulfur content in the PbS/CdS heterojunction.

#### 4.1.4. Copper Sulfide (CuS)

Copper sulfide is a binary inorganic material with the general formula Cu_x_S_y_ and one of the most important metal chalcogenides. It is a p-type semiconductor, with a band gap between 1.1 and 1.4 eV, and it has attracted great attention for implementation in solar cells over time, owing to its special properties and potential applications. One of these properties is the high electrocatalytic activity, which has established CuS as a typical counter electrode for quantum dot sensitized solar cells. It has also been proposed as a potential p-type absorber for thin-film solar cells and as an interfacial layer for PSCs, as well as a novel p-type semitransparent electrode [52]. A different set appears in the field of C electrode PSCs, though, where CuS has received much less attention and the reports of its use are only few, with all of them demonstrating the great potential of the material as a hole transport media, either solely or in combination with other materials.

Hu et al. demonstrated the enhanced hole transport properties of a C electrode, in which CuS nanostructures were employed as additives [102]. The nanostructures were prepared by a simple precipitation method and directly mixed with graphite and carbon black to form a slurry, which, after annealing at a low temperature, resulted in an ultra-low-cost hybrid counter electrode. The as-fabricated C-PSCs using the hybrid CuS-C electrode of the typical normal structure employing a compact and mesoporous TiO_2_ layer as the ETL and the CH_3_NH_3_PbI_3_ perovskite, with no additional hole transport layer used, exhibited significantly enhanced performance compared to the C-PSCs employing the bare C electrode. The increase in Voc from 0.93 to 0.98 V was explained by the valence band maximum being much lower than the C layer, while the high hole mobility of CuS in the hybrid electrode was described to produce increased Jsc values from 16.14 to 18.26 mA/cm^2^. Overall, the C-PSCs employing the hybrid CuS-C electrode exhibited a maximum PCE of 11.8%,while the C-PSCs employing the bare C electrode reached a maximum PCE of 9.36%.

In 2023, a novel approach was suggested by Liu et al. [103], where a composite of CuS–MXene was synthesized in situ by molecular self-assembly and exhibited excellent hole transport properties when applied in all-inorganic CsPbBr_3_-based C-PSCs (Figure 15a,b). MXenes have been on the forefront of research and application in PSCs for the past two years and have emerged as highly promising for trap state passivation. The authors combined the passivation properties of Ti_3_C_2_T_x_ MXene with the hole transport ability of CuS and demonstrated a CuS nanoparticle, modified MXene as an efficient hole extractor. The in-depth study of the CuS-MXene composite revealed that the W_F_ of MXene increases upon CuS addition, implying a p-doping effect and a favorable energy barrier reduction for the hole transport, which leads to a Voc enhancement. A larger built-in potential was recognized for the CuS–MXene-based PSCs, showing a stronger driving force for electron–hole pair separation and transport and a wider depletion region and lower charge accumulation at the PSC interface. The above confirmed the improved hole transport, while an interaction between CuS–MXene and CsPbBr_3_ was also identified, adding on to the improved interface contact, in addition to the passivation of defects that originate from the MXene. As a result, the planar C-PSCs with the FTO/SnO_2_/CsPbBr_3_/CuS–MXene/carbon device structure that were fabricated exhibited superior performance compared to the pristine devices and the devices employing only the MXene. The maximum PCE that was reached was 10.51%, with an impressive Voc of 1.629 V, a Jsc of 7.77 mA/cm^2^, and a high FF of 83.14% (Figure 14b). Moreover, the unencapsulated CsPbBr_3_ C-PSCs with the CuS–MXene interlayer retained 90% of their initial PCE after exposure in ambient air for 30 days.

#### 4.1.5. Other Structures

Apart from the aforementioned sulfides that are most commonly applied in PSCs in general, some alternative structures have also been reported to improve the performance of C-PSCs. These include nickel sulfide (NiS) and manganese sulfide (MnS), which have both been selected because of their hole transport properties and antimony sulfide (Sb_2_S_3_), which has been suggested as an electron transport layer.

Nickel sulfide was used to prepare a NiS-C composite by blending synthesized NiS with a commercial C paste at a proper ratio. The NiS that was synthesized possessed a bandgap of 3.1 eV, which is favorable for hole extraction, while the composite was found to be of high porosity. This composite was then applied as a dual-function hole transporting counter electrode in C-PSCs of the normal mesoporous structure, employing the CH_3_NH_3_PbI_3_ perovskite (FTO/c-TiO_2_/m-TiO_2_/perovskite/NiS-carbon composite). Even though the devices that use the composite electrode exhibited enhanced performance compared to the reference PSC, with the structure of FTO/compact-TiO_2_/porous-TiO_2_/perovskite/NiS/Cr/Pt-coated FTO, the PCE remained low (max 5.2%), mainly owing to the low Voc values (0.59 V) and FF (0.36) that the devices exhibited [104]. However, this approach is interesting and looks promising for application in the triple mesoscopic structured C-PSCs, considering the porous nature of the composite NiS-C electrode. Yu et al. also took advantage of the p-type properties of NiS and incorporated NiS nanoparticles as an interlayer between the inorganic CsPbI_2_Br perovskite and the C electrode [105]. The energy levels of NiS are well aligned with both the C electrode and the perovskite in use, facilitating the transport of holes and blocking the reverse transfer of electrons, therefore reducing interfacial recombination. The NiS nanoparticles also act as a passivation agent of surface defects, therefore reducing the trap assisted recombination, which, in turn, leads to an enhancement in the Voc values obtained (Figure 16). Overall, the inorganic C-PSCs bearing the NiS interlayer exhibit superior performance compared to the bare C-PSCs, with a significant improvement in the Voc values, reaching 1.32 V, as opposed to 1.23 V for the pristine device. The maximum PCE of the NiS C-PSCs was 14.36%, versus 12.09% for the pristine device, with a Jsc of 14.28 mA/cm^2^ and an FF of 76.17%, which is a considerable performance for all-inorganic, HTL-free C-PSCs.

An inorganic perovskite, CsPbBr_3_, was also employed by Li et al. [106] in C-PSCs employing MnS as an intermediate hole transporting layer. The perovskite was deposited using a thermal evaporation-assisted technique, which resulted in homogeneous, high-quality perovskite films. The all-inorganic devices exhibited PCEs as high as 10.45% versus 8.16% that the device free of an intermediate layer achieved as a result of both the optimum perovskite film morphology and the optimized band alignment between the perovskite and the C electrode after the MnS insertion, which favors the effective transfer of holes and limits recombination (Figure 17). Additionally, the C-PSCs with the MnS hole transporting layer exhibited enhanced stability, retaining 80% of its initial PCE after 100 days under stress conditions (80% RH, 85 °C) without any encapsulation.

Despite the well-known p-type properties of most metal sulfides, antimony sulfide (Sb_2_S_3_) is a semiconductor with high electron mobility (≈10 cm^2^ V^−1^ s^−1^). In order to investigate the possible positive effect of Sb_2_S_3_ on the electron extraction ability of the typical TiO_2_ ETL employed in PSCs and C-PSCs, Jing et al. [107] prepared Sb_2_S_3_ nanoparticles of about 20 nm using the hot injection method, which were uniformly distributed on the surface of TiO_2_ ETLs, forming a multilayered electron transport film. The composite TiO_2_@Sb_2_S_3_ exhibited higher conductivity than bare TiO_2_ after further modification with mercaptonic acid and enhanced electron extraction properties. Moreover, the inorganic CsPbI_2_Br perovskite films that were grown on the TiO_2_@Sb_2_S_3_ substrate presented higher crystallinity, lower roughness, and more uniform grains with more obvious grain boundaries. As a result, the PCE of C-PSCs that were prepared exhibited a maximum of 14.59%, which is a great enhancement compared to 12.29% for the devices prepared with the bare TiO_2_ as the ETL, owing mainly to an improvement in the FF values, which originated from the higher quality of perovskite crystals, and a slight improvement in the Voc obtained, which is as a result of faster electron extraction with less recombination at the ETL/perovskite interface.

To summarize, the sulfide chalcogenides that have been so far reported to have been implemented in C-PSCs are metal sulfides. Depending on the type of conductivity that metal sulfides present, they have been used mainly as charge transport layers and buffer layers in the appropriate positioning within the devices. In particular, MoS_2_, WS_2_, CuS, NiS, and MnS, as p-type materials, have achieved increased hole extraction properties when implemented in C-PSCs, as either intermediate or exclusive HTLs. On the other hand, CdS and Sb_2_S_3_, having a high electron mobility, have proven to be quite effective as an additional electron extracting layer, combined with typical ETLs, namely PCBM and TiO_2_. Regardless of the type of conductivity, all metal sulfides have been proven to perform well when adopted as passivation or buffer layers to increase the charge transport in the devices and passivate the defects, while a common and high-importance feature is that all of the devices that have been prepared, incorporating metal sulfides exhibit superior stability compared to the reference devices, which highlights the potential of metal sulfides as materials for achieving C-PSCs of high stability.

### 4.2. Selenides

Among the selenide chalcogenides, which have mostly been used in water splitting, transition metal selenides possess favorable properties, including high electrical conductivity and electrocatalytic activity and energy gap ~2 eV, which have led researchers to explore their potential application in PSCs. In PSCs with a metal electrode, tungsten diselenide (WSe_2_) has been employed as a hole transport layer and as a buffer layer at the perovskite/counter electrode interface, and molybdenum diselenide (MoSe_2_) has been reported mainly as a hole transport material. In PSCs with a C electrode, so far, only few reports exist, and the field has plenty of room for further exploration.

Zhou et al. studied the prospects of molybdenum diselenide (MoSe_2_) quantum dots to function as an effective additive to enhance the low-temperature TiO_2_ electron transport layer properties in HTL-free PSCs employing the inorganic CsPbBr_3_ perovskite and a low-temperature C electrode [93]. By incorporating the QDs in the ETL, they found that the doped L-TiO_2_:QDs films present a significant increase in their conductivity values under illumination and a simultaneous decrease in their electron trap-state densities contrary to the pristine TiO_2_ film, which shows negligible differences in both values. It was, therefore, implied that the doped ETLs have a high electron transfer ability, combined with reduced carrier recombination originating from effective trap passivation, which is expected to be more evident in the TiO_2_:MoSe_2_ composite ETL. Moreover, the implementation of transition metal dichalcogenides (TMDs) in the TiO_2_ was reported to have a beneficial effect on the crystallization of CsPbBr_3_ perovskite, leading to a high crystal quality with reduced defects, which further inhibits recombination, while a calculated upshift of the conduction band of the doped TiO_2_ ETLs signified enhanced interfacial electron extraction efficiency (Figure 18). The FTO/L-TiO_2_:MoSe_2_/CsPbBr_3_/C structure PSCs that were fabricated exhibited optimal performance, obtaining PCEs as high as 10.02% with an ultrahigh Voc of 1.615 V and a Jsc of 7.88 mA/cm^2^, which is much higher than the pristine TiO_2_, which exhibits a PCE of 8.25% with a Voc of 1.549 V and a Jsc of 6.63 mA/cm^2^.

Even though this is the only experimental report so far on the use of selenide chalcogenides in C electrode PSCs, a computational study by Ijaz et al. in 2023 [108] demonstrated the great potential of a series of inorganic electron transport materials (ETMs) for application in HTM-free PSCs based on the FTO/TiO_2_/CH_3_NH_3_PbI_3_/C structure. Among the materials studied, Zinc Selenide (ZnSe) was found to, by far, outperform the rest of the ETMs under study, exhibiting a Voc of 1.25 V, a Jsc of 24.77 mA/cm^2^, an FF of 86.29%, and a PCE of 26.76% of the simulated device after optimization of the ZnSe layer’s thickness and doping concentration (Figure 19).

To conclude, metal selenides have shown a theoretical, great potential as alternative ETLs; however, there still lacks enough experimental work to support the claim. This gap, though, can be filled in the future through deeper research in the application potentials of these materials.

### 4.3. Ternary

The broadest application of chalcogenides in PSCs, both in general and specifically PSCs with a C electrode, is related to the use of ternary chalcogenides, which are chalcogenides composed of a metal cation, a chalcogen anion, and a third cationic element. Ternary chalcogenides have an optical band gap that well matches the solar spectrum, significant photoluminescence properties, and considerable catalytic activity, while at the same time being sustainable and free of toxic elements. They are typically used as hole transport materials, given their p-type characteristic and their tunable energy level, which depends on their stoichiometry. There are several reports on the use of ternary chalcogenides in metal electrode PSCs, with the most commonly used chalcogenide type being the chalcogenide bearing the Cu metal. An enhancement of the hole transport properties of the as-prepared devices has been recorded in all cases, where the ternary chalcogenide has been paired with both hybrid organic–inorganic and all inorganic perovskites [109]. These positive findings have also been transferred to C-PSCs, where there are several reports of ternary chalcogenides being utilized as the hole transport layer, to increase the hole transport properties of the C electrode and to reduce the recombination of charges at the HTL/C interface or perovskite/C interface in the case of HTL-free devices.

#### 4.3.1. Copper Indium Sulfide (CuInS_2_-CIS)

Copper indium sulfide is the most widely reported chalcogenide that has been applied and studied in C-PSCs. The first report was in 2018 when Ding et al. [110] introduced the idea of applying an interlayer of eco-friendly (Cd-free and Pb-free) CuInS_2_/ZnS quantum dots on the film of the inorganic CsPbBr_3_ perovskite, in C-PSCs with the structure FTO/c-TiO_2_/m-TiO_2_/CsPbBr_3_/C, to tackle the interfacial charge recombination problem of all-inorganic C-PSCs, which is a consequence of the large energy difference (0.6 eV) between the inorganic perovskite and the C electrode. The size, the energy levels, and the bandgap of the CuInS_2_/ZnS QDs were tuned during synthesis through the nucleation temperature. Using this method, QDs with HOMO levels varying from −5.23 eV to −5.12 eV were obtained. The insertion of the QD interlayer has proven to be able to bridge the gap between the VB of the CsPbBr_3_ perovskite and the work function of the C electrode, thus promoting the successful transfer of holes and reducing the electron-hole recombination at the perovskite/C interface (Figure 20a). As a result, the C-PSCs that were fabricated, which included the CuInS_2_/ZnS QD interlayer, exhibited PCEs that were increasing, with the increasing bandgap of the QDs used, reaching a maximum value of 8.42%, while the PCE of the devices with no QD interlayer was limited to 6.01%. A significant enhancement in the Voc and FF values was expected as a result of the reduced recombination and enhanced charge transfer, while an increased value of Jsc was also noted (Figure 20b,c). Finally, the unencapsulated C-PSCs presented superior stability, maintaining 94% of the initial PCE even in high-humidity conditions.

In a similar work, the same group employed core/shell structured CuInS_2_/ZnS QDs, having a valence band of approximately −5.25 eV, as a hole transport layer in C-PSCs with the structure FTO/c-TiO_2_/m-TiO_2_/CsPbBr_3_/QDs/C [111]. After the optimization of the alkyl chain length of alkyl acid ligands ranging from 4 to 24 CH_2_ groups, with the optimum being 12 CH_2_ groups, a maximum efficiency of 9.32% was obtained for the QDs device, which was an improvement compared to 7.38% for the reference device. As expected, the charge transfer properties after the QD incorporation were enhanced, with a simultaneous minimizing of recombination, leading to improved Voc and FF values obtained for the QD devices over the HTL-free device. This work stands out for the incorporation of a long persistence phosphor (LPP) into the carbon ink used to prepare the cathode, with the scope of reabsorption or conversion of low-energy photons, to further improve the performance. Indeed, the addition of 10% of LLP materials, composed of Al, Sr, Y, Eu, S, and O elements, to the C12-QD-based C-PSCs were proven to broaden the light response without altering the physical and electrical properties of the C electrode, ending up in an enhancement of all electrical parameters, which resulted in the maximum PCE of 10.85% being achieved.

A novel approach was further proposed by the same group, where the moisture penetrating the C-PSCs was utilized by an all-inorganic PSC that simultaneously harvests solar and water-vapor energies [112]. The proposed device took advantage of both the ternary CuInS_2_/ZnS alloy quantum dots that were employed as the hole transport material at the CsPbBr_3_/carbon interface and graphene quantum dots (GQDs) that were used to modify the TiO_2_/CsPbBr_3_ interface. The maximum PCE that the bifunctional device exhibited was 9.43%, with a distinguishing stability of the electrical parameters, where 98% of the initial PCE was retained after 40 days.

Teimouri et al. reported on the use of CuInS_2_ as an efficient hole transport layer in novel C-PSCs of the planar FTO/SnO_2_/perovskite/CIS/C structure, where the Cs_0.05_(MA_0.17_FA_0.83_)_0.95_Pb(I_0.83_Br_0.17_)_3_ mixed cation–mixed halide perovskite was grown using the electrochemical deposition method [113]. Their approach yielded devices that achieved PCEs as high as 10.16%, not much inferior to 13.02% for the spin-coated device, by using a method that is simple, low-cost, and scalable. The same group very recently reported on C-PSCs with the same planar structure, employing CuInS_2_ as the hole transport layer, combined with the Cs_0.05_(MA_0.17_FA_0.83_)_0.95_Pb(I_0.83_Br_0.17_)_3_ mixed cation–mixed halide perovskite that achieved PCEs as high as 16.5% [114]. The novelty of this work is in the treatment of the SnO_2_ electron transport layer with urea, which was proven to improve the conductivity of the ETL and to increase its wettability and roughness, which promotes the growth of perovskite crystals with superior crystallinity, yielding a superior PCE compared to 14.04% of the pristine SnO_2_ ETL-based devices.

The successful implementation of CuInS_2_ as the hole transport layer was also reported by Hoseinpur et al. [115]. In their work, the authors prepared a multilayered compact TiO_2_/SnO_2_/mesoporous TiO_2_ electron transport layer, which was used in C-PSCs employing the CH_3_NH_3_PbI_3_ perovskite and a low-temperature C electrode. By optimizing the thickness of the SnO_2_ layer, the authors increased the PCE of the resulting C-PSCs from 7.19% for the devices with no SnO_2_ layer to 13.14%. Moreover, after the insertion of the thin SnO_2_ layer in the C-PSCs, the devices showed an improvement in their stability.

On the other hand, Noori et al. [116] conducted a study on the optimization of the TiO_2_ electron transport layer thickness by varying the concentration of TiO_2_ in the ETL precursor solution. The solutions were then used to prepare the ETL in C-PSCs with the structure of FTO/compact-TiO_2_/mesoporous-TiO_2_/CH_3_NH_3_PbI_3_/CuInS_2_/C, which were fully ambient-processed. The optimum 20% wt. concentration resulted in C-PSCs with a maximum PCE of 13.09% and an average of 12.44%.

In order to replace the typical high temperature annealing process for the TiO_2_ electron transport layer, Zamanpour et al. proposed a fast light-curing procedure for application in PSCs employing CuInS_2_ as a hole transport material, Cs_0.05_(MA_0.17_FA_0.83_)_0.95_Pb(I_0.83_Br_0.17_)_3_ as the perovskite and low-temperature C electrode [117]. Their method is proposed as being low-cost, fast, energy-saving, and applicable in the large scale, being alternative for the fabrication of high-efficiency and low-cost PSCs. The light sources that were used were halogen-tungsten lamp (H-lamp, 1 kW) and a mercury lamp (M-lamp, 400 W), and their results revealed the capability of a halogen-tungsten lamp to effectively promote the sintering of the TiO_2_ layer in a comparable degree to the conventional sintering at 500 °C. The authors found that both lamps’ intensity was enough to decompose the ethyl cellulose used as a binder in the mesoporous TiO_2_ precursor paste, and the optimal curing times were found to be 5 min for the H-lamp and 20 min for the M-lamp. By using a light-curing method with the optimal conditions, for both the compact and the mesoporous layers, the authors were able to achieve C-PSCs that exhibited PCEs as high as 16.3%, which is almost equal to the PCE obtained by the high-temperature furnace sintering process (Figure 21). Their method, combined with the low-cost and printable CuInS_2_ hole transport layer and C electrode, can be a significant step towards high-efficiency, scalable, and low-cost C-PSCs.

Mahmoodpour et al. highlighted the capability of the CuInS_2_ hole transport layer to be applied through slot-die coating in fully printed C-PSCs, employing the Cs_0.05_(MA_0.17_FA_0.83_)_0.95_Pb(I_0.83_Br_0.17_)_3_ mixed-cation mixed-halide perovskite [118]. After adjusting the printing properties including substrate and print head temperature, printing speed, meniscus height, shim thickness, and ink injection flow rate, the authors optimized the layer thickness and quality, achieving a final 9.93% of PCE in small-area devices, which is not far from the 11.38% obtained from the corresponding spin-coated devices. Their results have a great significance, since they provide a guide towards printable C-PSCs with a method that can be directly applied to fabricate large-area C-PSCs, favoring the commercialization of the particular PSC structure with the C electrode.

Working towards the same direction and, in particular, towards the enhancement of C-PSCs stability for their potential future commercialization, Baghestani et al. proposed a novel composite material, in particular a conductive adhesive ink comprising a polymer, carbon black, and CuInS_2_ nanoparticles, which was used to wet the carbon foil in carbon-laminated PSCs [119]. After the wetting, an interfacial adhesive layer was created at the underlying CuInS_2_ hole transport layer and the carbon electrode, which allows for the C electrode to be applied by the simple press transfer method (Figure 22).

The devices that were prepared employed the mixed cation, mixed halide perovskite, and after an optimization of the ingredients as well as their proportions in the adhesive ink, they presented an outstanding 17.2% power conversion efficiency, which is comparable to Au-based PSCs (Figure 23a). At the same time, the carbon-laminated PSCs retained over 92% of their initial PCE over 54 days, which is a highly promising result (Figure 23b,c).

Very recently, in 2024, Kassem et al. took advantage of the favorable properties of both inorganic (CuInS_2_) and organic (polytriarylamine-PTAA) hole transport materials and prepared a composite organic–inorganic PTAA-doped CuInS_2_ hole transport layer [120]. The composite layer benefits from the high hole mobility, simple and low-cost synthetic process, and enhanced stability that the inorganic HTMs possess and, at the same time, from the electron-blocking and hole-transport properties, along with the stability of the organic PTAA. This CuInS_2_/PTAA binary HTL was applied in C-PSCs having the FTO/c-TiO_2_/m-TiO_2_/Cs_0.05_(MA_0.17_FA_0.83_)_0.95_Pb(I_0.83_Br_0.17_) mixed cations–halides perovskite/C structure. The authors proceeded to the optimization of the energy-level alignment of the composite HTL structure with respect to the number of layers per individual HTL and of the concentration of PTAA in the composite HTL. The binary HTL had a positive effect on all electrical parameters, and the optimized C-PSCs achieved a maximum PCE of 15.95% after 10 days of storage, while the corresponding PCE of the reference device with a single CuInS_2_ HTL was restricted to 14.68%. After an extensive study, this improvement was attributed to improved charge extraction at the HTL, reduced recombination, and passivation of trap states induced in the binary HTL, while, additionally, the PTAA can act as a barrier layer for the penetration of C nanoparticles to the absorber. The optimized C-PSCs also maintained ~70% of their initial PCE after 408 h of thermal stress under ambient conditions.

Taking for granted that CuInS_2_ is an established effective inorganic HTM that can be used in C-PSCs, the most recent reports are devoted to the further improvement of the C-PSCs that employ CuInS_2_ as the HTM. In particular, Hoseinpour et al. suggested the introduction of an interface passivation layer between the CH_3_NH_3_PbI_3_ (MAPbI_3_) perovskite and the CuInS_2_ (CIS) hole transporting layer (HTL) [121]. The surface passivation layer consisted of *p*-toluene sulfonamide (PTSA) in two concentrations (50 and 100 mg/mL) and was deposited with different numbers of layers, and the effect of the PTSA solution concentration and number of PTSA layers were investigated as per the performance of C-PSCs with the FTO/b-TiO_2_/m-TiO_2_/MAPbI_3_/PTSA-x-y/CIS/carbon (where m-TiO_2_ = mesoporous TiO_2_, x = PTSA solution concentration, y = number of PTSA layers) structure, prepared under ambient conditions. The optimized devices exhibited superior performance over the C-PSCs without the PTSA interlayer, which was attributed to improved charge carrier transfer and reduced hole-electron recombination, which boosted the performance of the PCE values to as high as 11.24% versus 6.85% for the reference.

A similar study was presented in 2023 by Heydari et al. [122]. The authors reported on the use of N,N′-di(naphthalene-1-yl)-N,N′-diphenyl-benzidine (NPB) p-type, small molecules to passivate the surface of perovskite films in C-PSCs using CuInS_2_ nanoparticles as the HTM. After the employment of the NPB passivation layer between the perovskite and the CuInS_2_ HTM, the surface defects were minimized and non-radiative recombination was decreased; at the same time, the charge carrier transfer time at the perovskite/HTM interface was accelerated. As a result, the devices that were subjected to this passivation strategy, with the optimum concentration of NPB being 5%, exhibited PCEs as high as 16.11% versus 14.92% for the device without the NPB passivation layer (Figure 24), mainly owing to the higher Jsc values obtained, as a result of the faster charge transfer and the slight improvement in the FF values due to reduced defects acting as recombination centers. In addition, the un-encapsulated NPB-treated devices exhibited higher shelf-life stability, with a drop of the initial PCE of only 8% after 4000 h, which is much higher than the reference devices, which suffered a 42% loss at the same time period (Figure 24c).

Finally, the latest report on C electrode PSCs employing CuInS_2_ as a hole transport layer was presented by Mohammadi et al. in a work that did not focus on the CIS HTL but delivered C-PSCs with a notable PCE as high as 18.5% [123]. This was achieved by employing brominated porphyrin as an additive in the Cs_0.05_(MA_0.17_FA_0.83_)_0.95_Pb(I_0.83_Br_0.17_)_3_ mixed halide perovskite, which was implemented in C-PSCs of the FTO/SnO_2_/perovskite/copper indium disulfide (CIS)/Carbon planar structure. The main improvement compared to the reference device was in the significant increase in the FF, which is a result of optimum surface smoothness and perovskite film uniformity, which reduces charge recombination and increases charge extraction and transmission capabilities. Additionally, as a result of reduced recombination and more effective charge transfer, the C-PSCs employing the perovskite with the porphyrin additive presented an increase in the obtained photocurrent density values, leading to an overall increase in the PCE (Figure 25).

Judging from the above reports, which are summarized in Table 3, we could say that overall, CuInS_2_ is a well-established, efficient and stable inorganic hole transport material that is widely used in C-PSCs that deliver high PCEs. Its consistent efficacy allows for the further exploration of novel materials and modifications in the rest of the C-PSCs’ components that can contribute to the further improvement of PSCs with a C electrode [123]. Moreover, having enough experimental data to support the capability of CuInS_2_ to serve as the HTL in C-PSCs, the challenge of this material’s application would be to proceed to the next step, which is its successful application and study in large-area devices.

#### 4.3.2. Other Structures

Even though the most widely applied ternary chalcogenide is CuInS_2_ and its alloys, there are some additional structures that have also been reported and yielded interesting results. One of the most promising materials that have achieved high PCE C-PSCs is Cu_2_SnS_3_ in the form of nanocrystals, which was proposed by Yu et al. as a hole transport layer, in C electrode PSCs of the planar FTO/SnO_2_/perovskite/HTL/C structure [124].

The authors were driven to investigate the potential of Cu_2_SnS_3_ as an inorganic HTL in C electrode PSCs both by the promising results that have been obtained by the incorporation of this material as an inorganic HTL in metal electrode devices, as well as its environmentally friendly nature, since it is composed of non-toxic and earth-abundant elements. The nanocrystals were prepared by the hot injection method and then applied by spin-coating on the FAPbI_3_ perovskite layer, as shown in Figure 26.

The authors achieved a uniform, compact Cu_2_SnS_3_ film, with favorable charge extraction and transfer properties, which are also reflected in the PCE of the corresponding devices, reaching a maximum of 16.75% (Figure 27). Moreover, the C-PSCs with the Cu_2_SnS_3_ HTL presented a high degree of reproducibility and remarkable stability, with the PCE remaining almost unchanged after 30 days of storage in ambient conditions.

In a different approach and taking advantage of the favorable properties of chalcogenide quantum dots, including a tunable bandgap, which is dependent on the size and determined by the synthetic process parameters, and high luminescence, Li et al. [125] prepared CdZnSe@ZnSe colloid alloy QDs, which were then implemented in all-inorganic C-PSCs of the FTO/c-TiO_2_/m-TiO_2_/CsPbBr_3_/QDs/carbon structure. The QDs were proven to improve the perovskite/C interface, leading to the enhancement of FF values obtained and an overall PCE as high as 8.65% for the C-PSCs employing QDs of a 2.16 eV bandgap, as opposed to 7.53% for the bare HTL-free devices. This improvement in the PCE values incorporating the QDs was also proven to arise from reduced bi-molecular charge recombination owing to the optimum energy alignment between the QDs and the C electrode, compared to the bare device, as shown in Figure 28.

Overall, the first attempts of less common types of ternary chalcogenides, which are CuSnS_3_ and CdZnSe@ZnSe, to be applied as the HTL in C-PSCs, have yielded positive results. This proves that there is still a great number of chalcogen and metal combinations that could be used to prepare chalcogenides that could be explored as possible candidates for demonstrating novel HTLs in C-PSCs, and further research on this type of chalcogenides could lead to unexpectedly positive results.

### 4.4. Quaternary

Quaternary chalcogenides are structurally complex chalcogenides that comprise four different elements, one of them from the chalcogen family, which presents a wide range of electrical, optical, and thermoelectric properties. Being structurally and chemically flexible, which makes them suitable for a wide range of applications, these materials have drawn a great amount of attention. The most widely applied in PV technology is CuInGaS(CIGS), which has been considered a promising alternative to silicon for thin-film solar cells owing to its suitable bandgap that has succeeded in yielding high PCEs.

Their favorable properties and their established efficacy as photovoltaic materials have driven the research community to explore their potential implementation in PSCs in order to take advantage of the materials’ chalcogenides and perovskites exquisite properties. In C electrode PSCs, the reports are still few and mainly concern Copper Indium Gallium Sulfide (CuInGaS-CIGS) and Copper Zinc Tin Sulfide (CuZnSnS-CZTS), but the interest in exploring more structures is rising.

#### 4.4.1. Copper Indium Gallium Sulfide (CuInGaS-CIGS)

The main function that CIGS offers in C electrode PSCs is that of the hole transport layer, even though it has also been successfully as the absorber, in thin-film solar cells. The potential and prospects of CIGS as an efficient hole transport material for C electrode PSCs were studied widely by the group of Behrouznejad et al. During their study, they prepared C-PSCs having the structure of FTO/c-TiO_2_/m-TiO_2_/perovskite/HTL/C, and they proceeded to the optimization of C paste composition, with respect to the type and ratio of carbonaceous materials, binders, and solvents, in order to achieve PCE comparable to the reference PSCs employing spiro-OMeTAD as the HTL and a gold (Au) electrode [126]. The weight percentage of carbon black to carbon black plus graphite that was examined was 30, 40, and 50%. The solvents that were used in the carbon paste were selected to be toluene and chlorobenzene, and the polymers that were investigated as binders were polystyrene (PS) and polymethylmethacrylate (PMMA). The perovskite that was used in this study was chosen to be the mixed cation–mixed halide Cs_0.05_(MA_0.17_FA_0.83_)_0.95_Pb(I_0.83_Br_0.17_)_3_, and the compound that was chosen to serve as the HTM was CuIn_0.75_Ga_0.25_S_2_. It was found that the type of polymer does not have significant effects on the resulting carbon paste morphology, contrary to the carbon black-to-carbon powder weight ratio. However, it has a great effect on the resistivity of the electrode, which was found to be 30–50% lower in the electrodes comprising PMMA, compared to PS. When applied as an electrode to PSCs, the C pastes deliver devices with PCEs ranging from 1% for the device employing spiro-OMeTAD and C to a maximum of 15.9% for the device with a C electrode comprising 12% C powder, 4% wt PMMA dissolved in CB, and CIGS as the HTM. This value is comparable to the PCE of 16.3% obtained for the reference FTO/TiO_2_/perovskite/spiro-OMeTAD/Au device. It should be highlighted at this point that fabrication was performed outside of the glovebox. These results indicate the highly promising capability of CIGS as an alternative HTM in C PSCs.

After establishing the efficacy of CIGS as a HTL for C electrode PSCs, the group proceeded to the variation of Indium-to-Gallium ratio in the CIGS chemical structure and the study of the effect it has on the hole transport ability of the material [127]. It is well known that any alteration in the stoichiometry of quaternary chalcogenides induces changes to the bandgap and results in great variations in the obtained photovoltaic parameters of the resulting solar cells. In order to study this effect in C-PSCs employing CIGS as the HTL, the group varied the composition of CuIn_x_Ga_(1−x)_S_2_, where x = 1, 0.75, 0.5, 0.25, and 0. They found that by decreasing the Ga-to-In atomic ratio, there was also a decrease in the bandgap energies, which varied from 2.34 eV for In = 0 to 1.49 for In = 1 in CuIn_x_Ga_1−x_S_2_. All the compositions were found to be suitable as HTMs, presenting similar values of calculated τ_1_ and τ_2_ (fast decay and slow decay lifetime) to the reference, established spiro-OMeTAD HTM. As per their photovoltaic performance when incorporated in C-PSCs, the formulation of In/Ga = 0.75/0.25 yielded the highest-performing device, with a PCE as high as 16.45% and an outstanding value of Voc equal to 1.1 V, which indicates a lower recombination rate and a fast hole extraction and transport with this particular material. Additionally, the CuIn_0.75_Ga_0.25_S_2_ formulation achieved the most uniform and high-quality film, which contributed to the lowering of charge transfer resistance and the better affinity with the C electrode, which led to a high FF value of 0.65. All the prepared devices showed high stability in their electrical parameters after 1 month of shelf storage, indicating that CIGS, regardless of the precise composition, can be used as an effective, inorganic, alternative HTM to spiro-OMeTAD.

Finally, after having demonstrated some promising results regarding the use of CIGS as an HTM, the group performed an in-depth study on the preparation of CuIn_0.75_Ga_0.25_S_2_ (CIGS) nanoparticle inks [128]. They focused on achieving stable inks, with the capability to deliver high-quality, uniform thin films. For this purpose, a variety of solvents were explored, including toluene, chloroform, ethanol, methanol, and their mixtures, with the inks prepared with solvents of polarity index ranging from 0.26 to 0.36 showing a good degree of stability. Moreover, chloroform exhibited the optimum film formation ability. The CIGS films that were prepared using the CF-based ink were used as the HTL in C-PSCs with the structure FTO/c-TiO_2_/m-TiO_2_/perovskite/CIGS/Carbon, where the perovskite used was the mixed cation–mixed halide Cs_0.05_(MA_0.83_-FA_0.17_)_0.95_Pb(Br_0.17_I_0.83_)_3_. The devices were able to deliver PCEs as high as 16.48%, proving, on the one hand, the capability of CIGS to effectively transfer holes and, on the other hand, the success of the authors’ method to deposit a uniform and efficient film using the optimized ink.

#### 4.4.2. Copper Zinc Tin Sulfide (CuZnSnS-CZTS)

Copper Zinc Tin Sulfide (CZTS) is a non-toxic and earth-abundant p-type semiconducting chalcogenide, with high thermal and chemical stability, low cost and facile synthesis methods, and a tunable bandgap. It has been most widely known for its efficacy as a light absorber in low-cost, thin-film solar cells, the “kesterite solar cells” that are widely studied. In perovskite solar cells, as most chalcogenides, CZTS has been adopted as the hole transport material owing to its high hole mobility, including devices with a C electrode.

The first report on the adoption of CZTS as the HTM in C-PSCs was presented in 2019 by Cao et al. [129]. Driven by the promising results that had been reported regarding the use of CZTS in Au electrode PSCs, the authors prepared CZTS nanoparticles using the hot injection method, which was further used to prepare thin HTL films in PSCs employing the CH_3_NH_3_PbI_3_ perovskite and a low-temperature, paintable C electrode (Figure 29). After an optimization of the thickness of the CZTS film and the annealing time, they achieved a maximum PCE of ~13%.

Subsequently, in 2020, the same group proceeded to the modification of CZTS nanoparticles with hexanethiol, which resulted in the enhanced hole extraction and transport properties of the corresponding HTL [130]. The ligand-modified HTL was implemented in C-PSCs of the planar FTO/SnO_2_/perovskite/CZTS/C structure, where two different perovskite compositions were studied. The reference device employed the CH_3_NH_3_PbI_3_ (MAPbI_3_) perovskite and bare CZTS nanoparticles as the HTM and yielded PCEs of up to 14.27%. This was improved to 16.62% after the incorporation of the ligand-modified CZTS nanoparticles in the same device architecture, while switching to the CH(NH_2_)_2_PbI_3_ (FAPbI_3_) perovskite further increased the PCE values to an impressive 17.71%.

In 2021, they advanced the use of CZTS in C-PSCs and presented a scalable, one-step heating-up method to synthesize gram-scale Cu_2_ZnSnS_4_, which was employed as the HTL in C-PSCs with the planar FTO/SnO_2_/perovskite/CZTS/C structure [131]. The perovskite used was CH_3_NH_3_PbI_3_ and the C-PSCs that was prepared with the CZTS nanoparticles synthesized with this method, after an optimization of the reaction parameters, achieved PCEs as high as 16.1% (Figure 30). This result is of great importance since it paves the way for the application of CIZTS as a hole transporter in large-area C-PSCs and in a large scale.

In a different approach, Mashreghi et al. presented a two-phase method to prepare Cu_2_ZnSnS_4_ nanoparticles with an average size of 9.5 nm [132]. In this method, contrary to the typical hot-injection method that is widely used for nanoparticle synthesis, simple precursors and non-hazardous organic solvents are used, making it a cheap and environmentally friendly alternative synthetic process. The as-fabricated nanoparticles were then deposited by spin-coating on the CH_3_NH_3_PbI_3_ perovskite layer to make the HTL in C-PSCs having the FTO/c-TiO_2_/m-TiO_2_/perovskite/CZTS/C configuration. The devices employing the kesterite (CZTS) HTL exhibited significantly enhanced photocurrent density and FF values compared to the HTL-free devices and a final PCE of 9.3% compared to 6.1% for the reference devices. However, the CZTS C-PSCs exhibited lower stability values, and after 96 days of shelf storage, their performance was lower than that of the HTL-free C-PSCs, which is a matter for improvement (Figure 31).

The most recent report on Cu_2_ZnSnS_4_ in C-PSCs is the study of Heidariramsheh et al. in 2021, where a complete investigation of the influence of the Zn:Sn ratio in CZTS on C-PSCs performance was performed [133]. The authors observed the correlation of the Zn:Sn ratio with the energy bandgap, where after an initial drop in the bandgap values from 1.42 to 1.36 eV when Zn:Sn changed from 0 to 0.9, a blue shift (increasing bandgap) was noted with increasing Zn:Sn ratio from Zn:Sn0.9 to Zn:Sn1.7. The authors attributed the initial value of 1.42 eV for Zn:Sn0 to a difference in the phase of the material, which changed from wurtzite of Zn:Sn0 to kesterite with increasing Zn content, in addition to the presence of a secondary phase of copper sulfide that also increases the bandgap values. The energy levels of the conduction band minimum (CBM) and the valence band maximum (VBM) of the synthesized CZTS NPs were used to construct the energy level alignment diagram, which demonstrated that the increasing Zn amount in the CZTS promoted a favorable shift and a better match with the energy levels of the mixed cation, mixed halide Cs_0.05_(MA0.83FA_0.17_)_0.95_Pb(Br_0.17_I_0.83_)_3_ perovskite that was used, suggesting a better hole extraction and reduced recombination of charges. Regarding the conductivity of the films, an initial increase in the Zn:Sn ratio from 0 to 0.9 and then 1.1 induced a decrease in the electrical conductivity, which was followed by an increase in conductivity with a further increase in the Zn content (Figure 32), also giving a clue on the corresponding best-performing final device.

Indeed, the variation in the Zn content resulted in C-PSCs with significantly different performances, with prominent variations in the Voc values, which is a clear indication of the difference in hole extraction capability and the different recombination kinetics with different CZTS compositions. The highest-performing device achieved PCEs as high as 15.49%, and it was at a Zn:Sn ratio of 1.5 (Figure 33). This study provided a significant contribution to the understanding of the mechanisms for efficient hole extraction and transfer by the inorganic Cu_2_ZnSnS_4_ HTM and has a significant contribution to the advancement of research in this topic.

#### 4.4.3. Other Structures

Because of the advantage of tunability on request and by changing their composition, besides CuInGaS and CuZnSnS, which have been applied the most, among quaternary chalcogenides, recent work has proposed the adoption of additional structures that could replace the organic HTMs in C-PSCs, and interesting results have been obtained.

Li et al. synthesized and applied multilayered films of Ag-In-Ga-S quantum dots as HTL in all-inorganic CsPbBr_3_ perovskite-based C-PSCs, which achieved noteworthy Voc values, exceeding 1.4 V [134], while Cheng et al. demonstrated an outstanding PCE of 18.02% in C-PSCs of the planar FTO/SnO_2_/Cs_0.05_(MA_0.83_FA_0.17_)_0.95_Pb(Br_0.17_I_0.83_)_3_/HTM/C structure with the novel Cu_2_ZnGeS_4_ HTM proposed [135]. The motivation for this work was the previous experience of the group in a successful incorporation of Cu_2_ZnSnS_4_ as the HTM in C-PSCs [130] and the observation that the low band gap of CZTS could result in far-from-optimum band alignment with the perovskite and a high charge carrier recombination rate at the perovskite/HTL interface. Instead, the replacement of Sn with Ge resulted in a quaternary chalcogenide with a bandgap of ~2.0 eV compared to the 1.5 eV for CZTS, which could have a better energy band alignment with the perovskite, favoring the extraction and transfer of holes in a more effective way. Indeed, the devices prepared with the novel CZGS HTL exhibited high FF and Voc values, which led to high-performing solar cells with high reproducibility and stability (Figure 34).

Finally, nanoparticles with the structures of Cu_2_AgInS_4_ and Cu_2_AgInSe_4_ [136], as well as a composition of mixed Cu_2_AgIn(S_0.5_Se_0.5_)_4_ [137], were also explored as potential HTMs for C-PSCs by the group of Angaiah et al. However, the corresponding device efficiencies still remain low, with the maximum achieved being 4.64% for Cu_2_AgIn(S_0.5_Se_0.5_)_4_. These reports, though, suggest that there is still plenty of room for compositional engineering to achieve novel HTMs for C-PSCs based on quaternary chalcogenides.

In summary, quaternary chalcogenides, in a variety of structures, have proven to be highly efficient as inorganic HTLs in C-PSCs and a promising alternative to the organic HTLs used so far. The most well-studied structures are CuIn_x_Ga_1−x_S_2_ and Cu_2_ZnSnS_4_, which have been extensively investigated by a limited number of groups. Due to the complex nature of these structures, which have a determining effect on the resulting material properties, only a few variations of quaternary chalcogenides have been reported so far, implying that the complex structure and the complex preparation method required for these materials are a drawback towards their greater application by many groups, which could lead to the optimization of structures and fabrication methods that would then create potential for their implementation in large-area devices.

## 5. Prospects and Challenges

Being a family of compounds of a large range, chalcogenides possess the unique advantage of tailorable properties according to the desirable application. Therefore, they constitute a highly promising class of materials for application in optoelectronics and, in particular, PSCs. From electron transport materials to absorbers, hole transport media, and even hybrid counter electrodes, chalcogenides have exhibited significant results in all devices where they have been applied so far, achieving improvement in the PCE values, as well as the stability of the resulting solar cells. Additionally, they can be environmentally friendly, non-toxic, and low-cost materials that can be deposited with a variety of methods in a wide range of solar cell structures.

Even though the reports on the use of chalcogenides are many and steadily rising in PSCs of the typical structure with a metal back electrode, these reports are significantly fewer when observing the trends in PSCs with a C electrode. It is a fact that any novel materials are first introduced and optimized in PSCs with a metal electrode, and afterwards, they are modified to meet the criteria and peculiarities that C-PSCs demand.

After having established the compatibility and efficacy of chalcogenides in metal electrode PSCs, now is the era where the positive effects of these materials can be transferred to the C-PSCs configuration as well. In fact, the reports of chalcogenide implementation in C-PSCs so far are related to their application only in the low-temperature C electrode configuration, whereas there are no reports yet in their utilization in C-PSCs of the high-temperature C electrode configuration, also known as the triple mesoscopic structure. The implementation of chalcogenides in this structure as well would be of high scientific, practical, and financial interests, considering that this structure is highly compatible with scalable and low-cost fabrication methods, such as screen and inkjet printing, which can be directly applied in industry for the production of large-area and large-scale application of PSCs and have already demonstrated significant results in HTL-free C-PSCs with high efficiency [11]. The implementation of chalcogenides in this structure can boost their efficiency, which, combined with the exceptional stability that this structure is characterized by, even in non-sealed devices, can lead to superior PSCs with performance comparable to the most widely applied thin-film solar panel technologies, such as polycrystalline Si, with a considerably lower cost. This advancement can highly contribute to the commercialization of PSCs, which is now hindered by their high cost, high toxicity, and low stability, making the technology unviable for further investments and wide application.

The challenges that need to be addressed in order for chalcogenides to be eligible for application in C-PSCs, which have particular structural and several constructing peculiarities compared to the metal electrode PSCs, are related to the tailoring of the chemical structure of the compounds in order to, on the one hand, achieve the appropriate size of the compounds so they can penetrate the charge transfer layers and the C electrode effectively as well as achieve homogeneous films with a full coverage. On the other hand, the chemical structure needs to be adjusted to achieve the optimum energy band alignment. Both of the above are crucial to mitigate the large recombination phenomenon that is observed in C-PSCs. Moreover, there are now enough reports indicating the beneficial effects of chalcogenides in C-PSCs, showing that the materials need to move to the next level and prove their efficacy in large-area devices.

Specifically, even though more reports on simple structured metal chalcogenides (i.e., binary chalcogenides) are related to the use of MoS_2_ as an interlayer and as a sole HTL, and despite the promising results obtained, these are of less value if not applied to both the LT- and HT-structure C-PSCs and then transferred to large-area devices and evaluated as per their performance and stability under the ISOS protocols. The results of such study and a comparison with the corresponding results that have been already been obtained in metal electrode PSCs [138,139,140] would be of a high interest and necessitate a consideration for further research and development for a possible product.

The same gap appears in the case of CuInS_2_, which has been widely reported and established as a highly efficient HTL in C-PSCs among ternary chalcogenides. Various modifications have been made to increase the device performance; however, there are still no attempts reported regarding the implementation of the material in large-area C-PSCs. The study of this material in lab-scale devices is mature enough for the scalability potential to be evaluated in order for further research to have a point. On the other hand, the investigation of a series of unexplored combinations of chalcogens and metals during the synthesis of ternary chalcogenides, as well as the ratio between them, could lead to unexpectedly positive results and would be of a high interest to attempt.

As for quaternary chalcogenides, in our point of view, their complex structure and the equally complex and costly preparation and deposition methods will constantly hinder the wide application of these materials, considering that any inconsistency in the structure defines the optoelectronic properties of quaternary chalcogenides and has a detrimental effect on the corresponding device performance. A method to surpass this issue and take advantage of their beneficial properties, low-cost, reproducibility, and scalable synthesis, as well as a deposition method, need to be developed, which could give a boost to this material’s potential.

Overall, the study of chalcogenides in PSCs has just started to become more intensive, and there is plenty of room for chalcogenides’ application in PSCs in general and even more space for investigation in PSCs with a C electrode, making chalcogenide structures the next star in material science for optoelectronic and energy harvesting applications

## 6. Conclusions

Chalcogenides are a part of a broad family of materials, which are distinguished by tunable bandgaps and excellent optical and electrical properties. These properties, along with the extensive research on their use in thin-film solar cells, make them both interesting and highly promising for incorporation in perovskite solar cells with a carbon electrode (C-PSCs).

So far, the most commonly applied compounds are transition metal chalcogenides (TMDs), with most reports focusing on Molybdenum Disulfide (MoS_2_), which has long been used in PSCs with a metal electrode, as well as OPVs, yielding promising results. In C-PSCs, there are reports on the use of MoS_2_ as a passivation layer, as an HTL, and as an additive in all ETL, perovskite, and C electrode constituents, mainly in all-inorganic C-PSCs that have achieved a maximum PCE of 10.13%. Tungsten Disulfide (WS_2_), on the other hand, has much fewer reports, with its use being restricted to a buffer layer between the charge transport layer/perovskite interface and the maximum PCE obtained approaching 16%, while there is only one report on Molybdenum Diselenide (MoSe_2_) as an additive in the ETL of C-PSCs that have achieved a notable Voc of 1.615 V.

Among the metal chalcogenide semiconductors, few reports exist on the use of Cadmium Sulfide (CdS), owing to its high electron mobility and Copper Sulfide (CuS) as an established HTM, while the use of other metal sulfides in C-PSCs has been scarce.

The most promising results, though, have been obtained from the application of ternary and quaternary chalcogenides, mainly as hole transporters in C-PSCs. In particular, Copper Indium Sulfide (CuInS_2_-CIS) is the most widely reported chalcogenide used in C-PSCs and has achieved PCEs as high as 18.5% in devices where it served as the HTL, making CuInS_2_ a well-established, efficient, and stable inorganic hole transport material and its further exploration almost mandatory. By incorporating Ga in the chemical structure, the quaternary Copper Indium Gallium Sulfide (CuInGaS-CIGS) chalcogenide has been obtained, which is also characterized by an established high hole transport capability, which has yielded C-PSCs with PCEs exceeding 16%. By alloying Sn with Zn, and creating the quaternary Copper Zinc Tin Sulfide (CuZnSnS-CZTS) chalcogenide, an equivalently efficient hole transport material is revealed, whose application as HTL in C-PSCs has yielded devices with PCEs as high as 17.71%. There are still few promising reports on the implementation of ternary and quaternary chalcogenides in C-PSCs; however, these results are only indicative, and this field requires further exploration, with a certainty that more significant results will be obtained.

The gathering and presentation of the reports that exist so far in the literature, which show that all chalcogenides have a positive effect on C-PSCs performance and stability, regardless of the chalcogenide type and structure, have the intention to trigger the interest in further investigations, which can lead to the resolution of the drawbacks that exist. One of them is related to the chemical tailoring of the most promising structures to achieve materials that will have optimum optoelectronic properties. Another issue that needs to be tackled is the deposition method of chalcogenides, which needs to be low-cost, scalable, and reproducible, as well as possible to be applied under ambient conditions using the most advanced industrial methods (e.g., slot-die coating, blade coating, spray coating, printing) [141], since scalability, besides efficiency, is an important matter to be considered when addressing perovskite solar cells in general.

To summarize, the field of chalcogenides in perovskite solar cells is of growing interest, and from this scientific trend, the carbon electrode perovskite solar cells could not be left behind. The first reports and investigations of chalcogenides in C-PSCs exhibited their compatibility and efficacy in a variety of roles in the device architecture, achieving high PCE values and considerable stability. Taking for granted the positive effects of chalcogenides in C-PSCs and considering that chalcogenides will gradually be on the forefront as potential materials for high efficiency, high stability, low cost, and environmentally friendly PSCs, further exploration of their application in C-PSCs and a step forward towards their application and study in large-area C-PSCs, which are the most commercially viable architecture among PSCs, are required and crucial to deliver significant results that can push the technology closer to its broader application.

## Figures and Tables

**Figure 1 nanomaterials-14-01783-f001:**
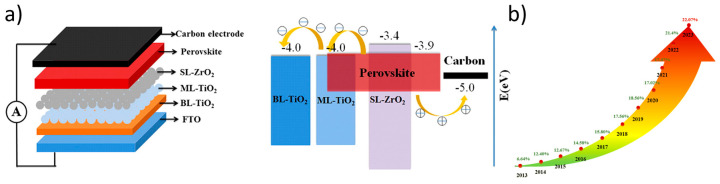
(**a**) The structure and energy diagrams of a carbon counter electrode perovskite solar cell [12]; (**b**) the evolution of the power conversion efficiency of carbon counter electrode perovskite solar cells, from 2013 to 2023 [13].

**Figure 2 nanomaterials-14-01783-f002:**
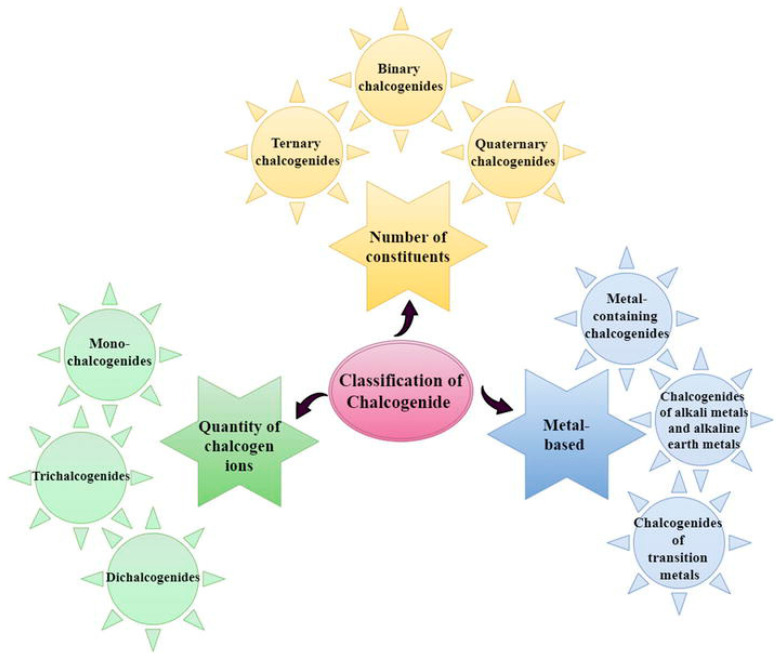
Classification of chalcogenides [18] (Reprinted from Ref. [18] DOI:10.5772/intechopen.1005357, 2023, Priya and Sagadevan, under Creative Commons Attribution 3.0 License).

**Figure 3 nanomaterials-14-01783-f003:**
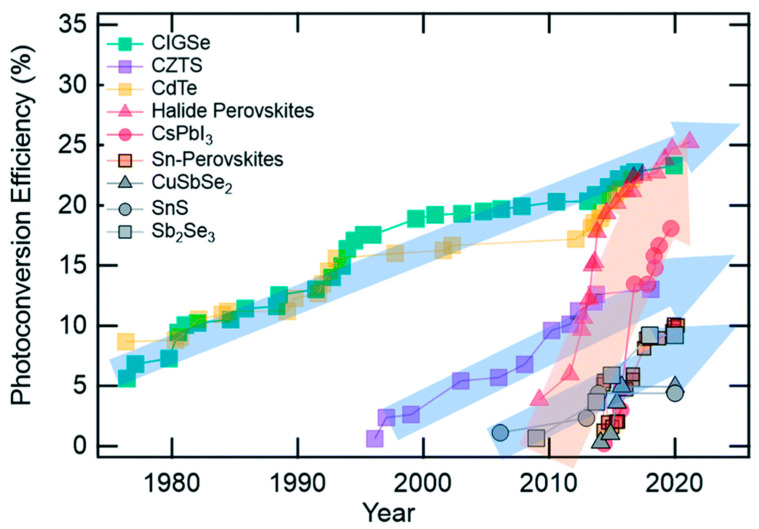
Evolution of cell performance for a variety of thin-film photovoltaic materials [74] (Reproduced from Ref. [74] doi:10.1039/D2FD00085G with permission from the Royal Society of Chemistry).

**Figure 4 nanomaterials-14-01783-f004:**
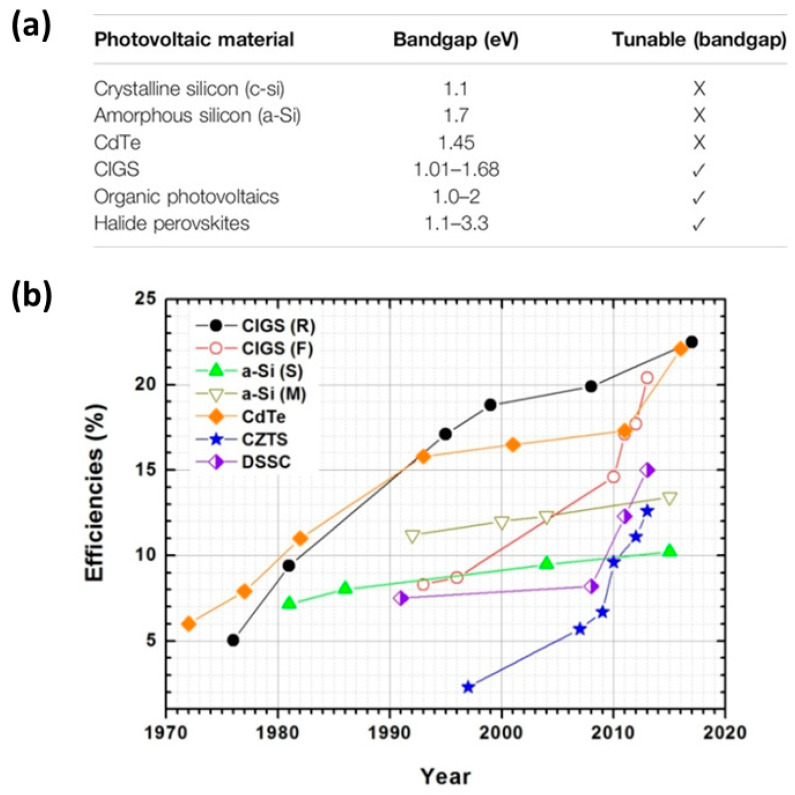
(**a**) Different photovoltaic materials and their bandgaps [78] (Reproduced from Ref. [78] doi:10.3389/fchem.2021.632021 under the terms of the Creative Commons Attribution License (CC BY)); (**b**) Historic record efficiencies of major types of thin-film solar cells (lab scale) [79].

**Figure 5 nanomaterials-14-01783-f005:**
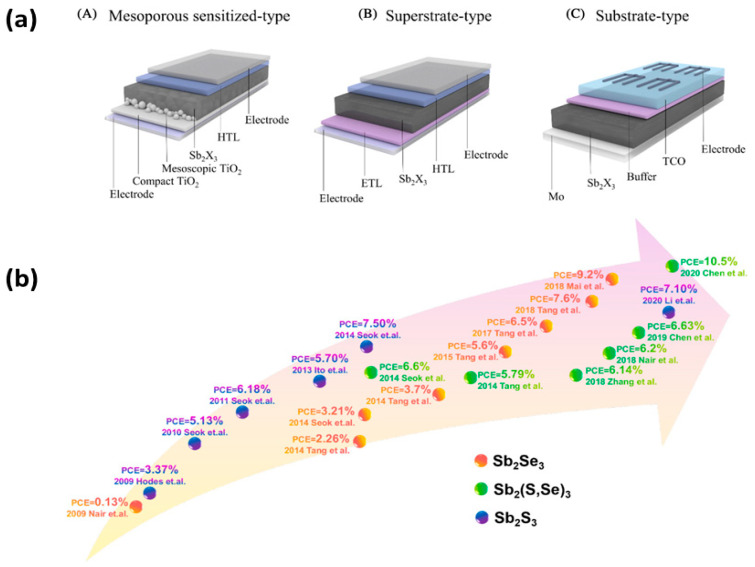
(**a**) Antimony chalcogenide solar cell configurations: (A), illustration of mesoporous sensitized-type antimony chalcogenide-based solar cells; (B), superstrate planar structure chalcogenide-based solar cells; and (C), a substrate planar structure chalcogenide-based solar cell. (**b**) The main achievements in antimony chalcogenide solar cells [81] (reproduced from Ref. [81] doi:10.1002/nano.202000288, 2021, Dong, Liu, wang and Zhang, under the terms of the Creative Commons CC BY license).

**Figure 6 nanomaterials-14-01783-f006:**
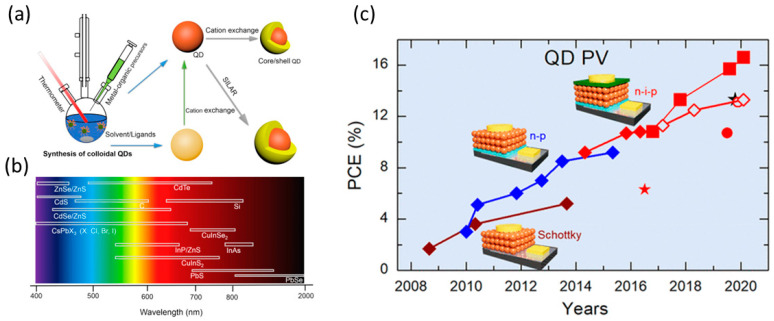
(**a**) Schematic representation of the synthesis of colloidal QDs via various approaches and their emission range. A synthetic apparatus used in the preparation of various structured QDs. The bare QDs can be synthesized via a cation exchange approach. The core-shell structure can be obtained via both a cation exchange approach and SILAR approach; (**b**) The emission range for representative QDs [82]; (**c**) PCEs for quantum dot photovoltaics (QD PV) over the years [83] (reprinted with permission from ACS Energy Lett. 2020, 5, 9, 3069–3100, Copyright 2020 American Chemical Society).

**Figure 7 nanomaterials-14-01783-f007:**
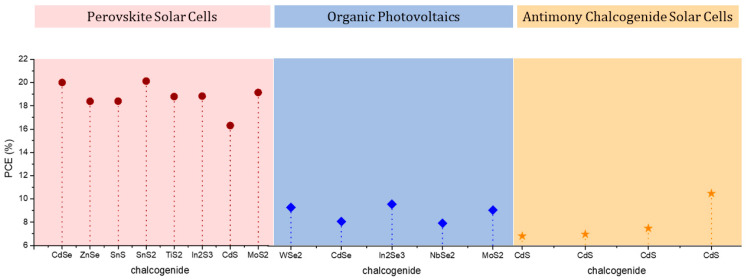
A summary graph with some representative PCEs obtained with various chalcogenides, employed as ETMs, in 3 types of solar cells.

**Figure 8 nanomaterials-14-01783-f008:**
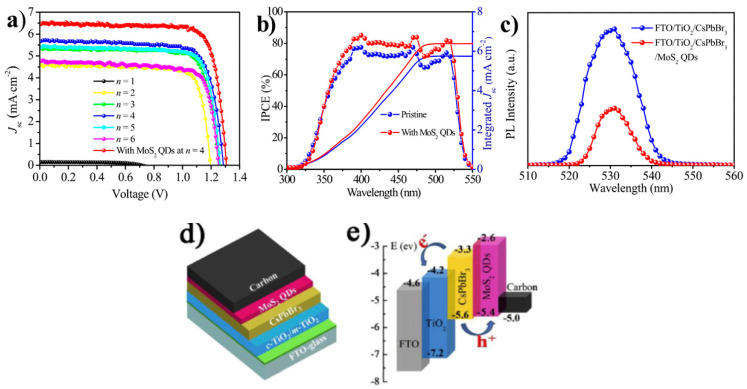
(**a**) J–V curves of the all-inorganic PSCs based on spraying-assisted deposition; (**b**) IPCE spectra and integrated current densities of the PSCs with and without MoS_2_ QDs; (**c**) PL emission spectra; (**d**) Schematic diagrams of an inorganic PSC; and (**e**) charge transportation processes [92] (reprinted with permission from ACS Appl. Mater. Interfaces 2023, 15, 48, 55895–55902. Copyright 2023, American Chemical Society).

**Figure 9 nanomaterials-14-01783-f009:**
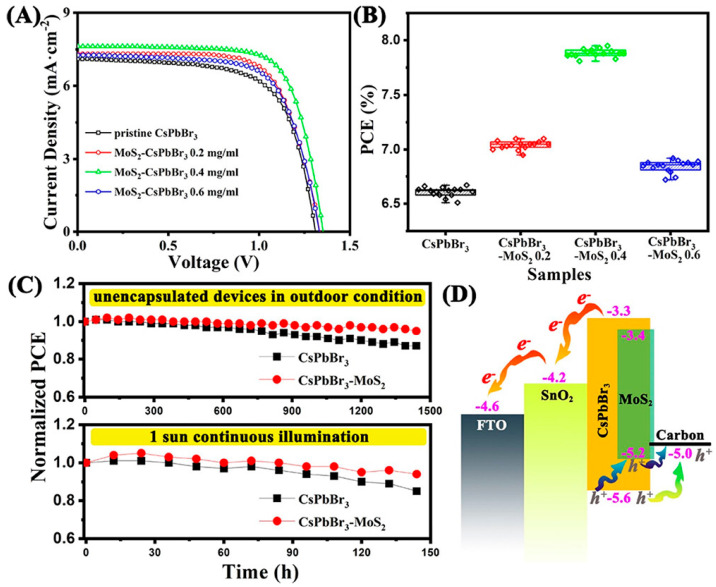
(**A**) J–V curves and (**B**) box-line graphs of C-PSCs (1 cm^2^) based on the absorber of CsPbBr_3_ and CsPbBr_3_–MoS_2_ with MoS_2_ concentrations of 0.2, 0.4, and 0.6 mg/mL; (**B**) variation of PCE of C-PSCs with time based on the absorber of CsPbBr_3_ and CsPbBr_3_–MoS_2_ (0.4 mg/mL) in outdoor conditions and under 1 sun continuous illumination; (**C**) schematic diagram of energy level alignment and (**D**) of C-PSCs with the structure of FTO/SnO_2_/CsPbBr_3_–MoS_2_/C [94].

**Figure 10 nanomaterials-14-01783-f010:**
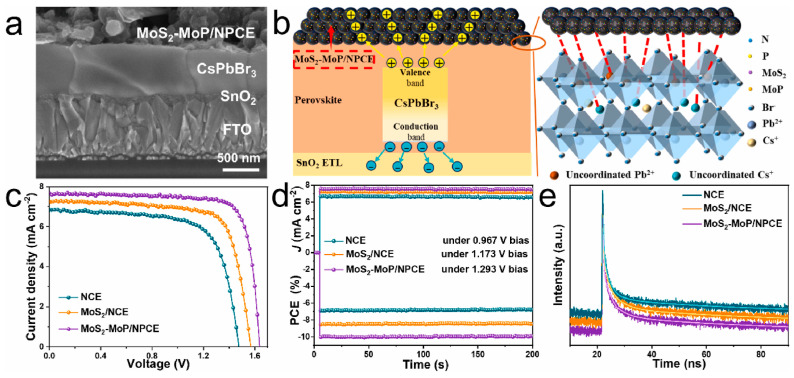
(**a**) Cross-sectional SEM image of MoS_2_–MoP/NPCE assembled HTL-free CsPbBr_3_ PSCs; (**b**) Schematic diagram of the function of MoS_2_–MoP/NPCE; (**c**) J–V curves and (**d**) steady-state output curves of HTL-free CsPbBr_3_ PSCs based on various carbon electrodes; (**e**) TRPL spectra of Glass/CsPbBr_3_/various carbon electrodes [95].

**Figure 11 nanomaterials-14-01783-f011:**
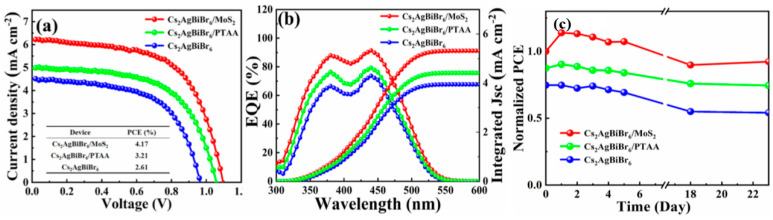
(**a**) J–V curves for the best-performing solar cells (inset table: maximum PCE values of three devices); (**b**) The EQE/integrated current densities and (**c**) PCE stability of three different devices [96].

**Figure 12 nanomaterials-14-01783-f012:**
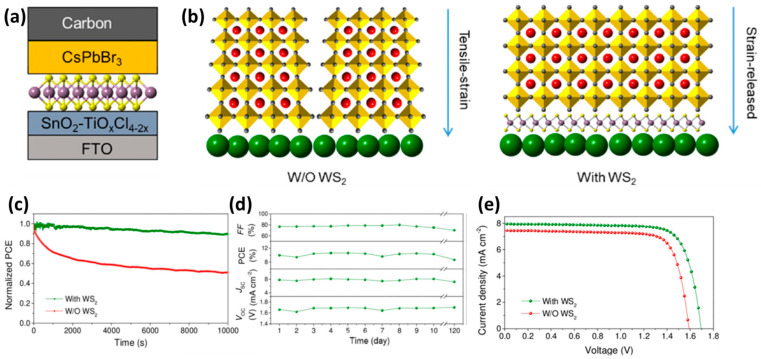
(**a**) Schematic diagram of an all-inorganic CsPbBr_3_ PSC; (**b**) Schematic diagram of residual strain distribution in CsPbBr_3_ grains with and without WS_2_; (**c**) Photostability of PSC devices with and without WS_2_ interlayer under one sun illumination; (**d**) Voc, Jsc, PCE, and FF stability of the device with a WS_2_ interlayer free of encapsulation at 25 °C and 80% humidity interlayer; (**e**) J–V curves of PSCs with and without WS_2_ interlayer [97].

**Figure 13 nanomaterials-14-01783-f013:**
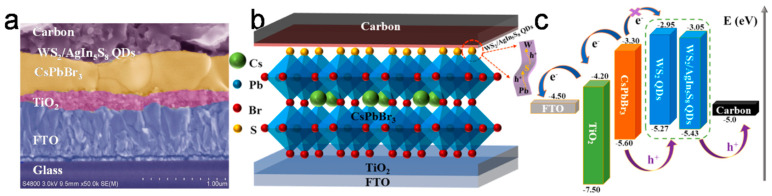
(**a**) A cross-sectional SEM image of CsPbBr_3_ PSCs; (**b**) Schematic diagram of WS_2_/AgIn_5_S_8_ QDs-based CsPbBr_3_ PSCs; (**c**) Energy level diagram of the charge transfer process in various CsPbBr_3_ PSCs [98].

**Figure 14 nanomaterials-14-01783-f014:**
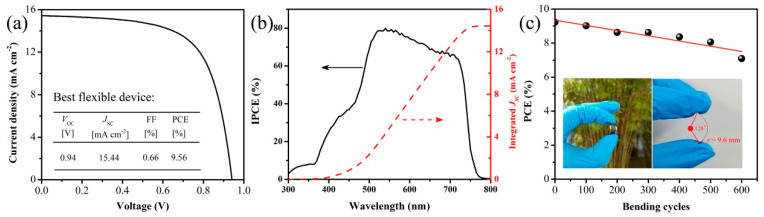
(**a**) J–V characteristic measured under the AM 1.5G simulated solar light; (**b**) the corresponding IPCE spectra for the best-performing PEN/ITO/CdS/PCBM/Perovskite/CuPc/C flexible device; (**c**) PCEs of flexible PSC for bending cycles. The inset shows the flexible device’s photograph [100].

**Figure 15 nanomaterials-14-01783-f015:**
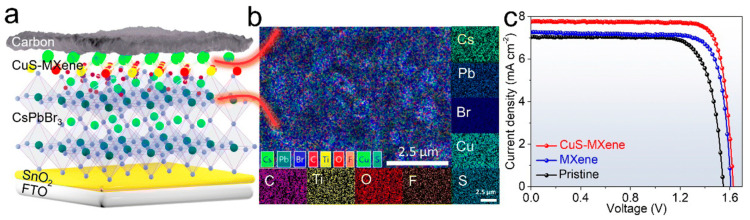
(**a**) Schematic diagram of CuS–MXene-based all-inorganic CsPbBr_3_ PSCs; (**b**) EDS elemental mapping images of the CsPbBr_3_/CuS–MXene film; (**c**) The J–V curves of various PSCs [103].

**Figure 16 nanomaterials-14-01783-f016:**
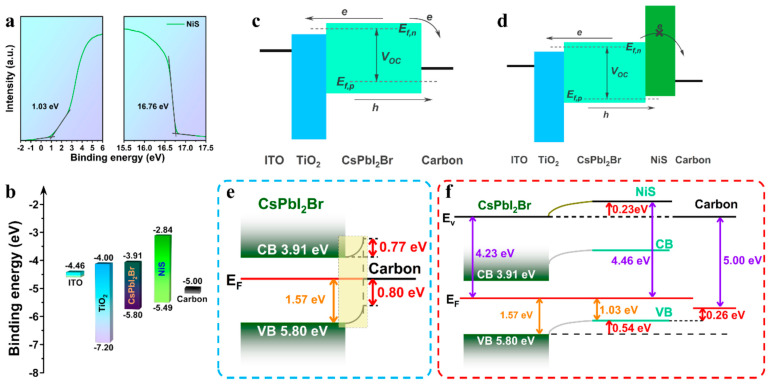
(**a**) UPS spectra of the NiS NPs layer; (**b**) Energy level alignment of the functional layers in the CsPbI_2_Br PSCs with NiS modification. The schematic diagram of quasi-Fermi level splitting (**c**) of pristine CsPbI_2_Br PSC and (**d**) of CsPbI_2_Br/NiS PSC; (**e**) schematic diagrams of band bending of CsPbI_2_Br/Carbon and (**f**) CsPbI_2_Br/NiS/Carbon interfaces [105].

**Figure 17 nanomaterials-14-01783-f017:**
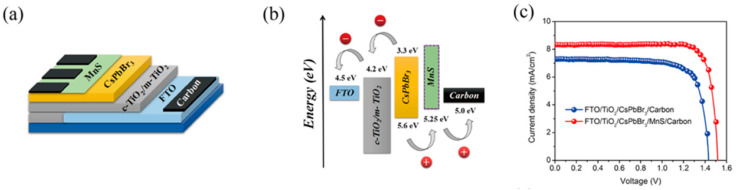
(**a**) Schematic view of the inorganic cell structure; (**b**) Energy-level diagram of the PSC; (**c**) J–V curves of inorganic PSCs with and without the MnS intermediate layer [106] (reprinted with permission from ACS Appl. Mater. Interfaces 2019, 11, 33, 29746–29752. Copyright 2019 American Chemical Society).

**Figure 18 nanomaterials-14-01783-f018:**
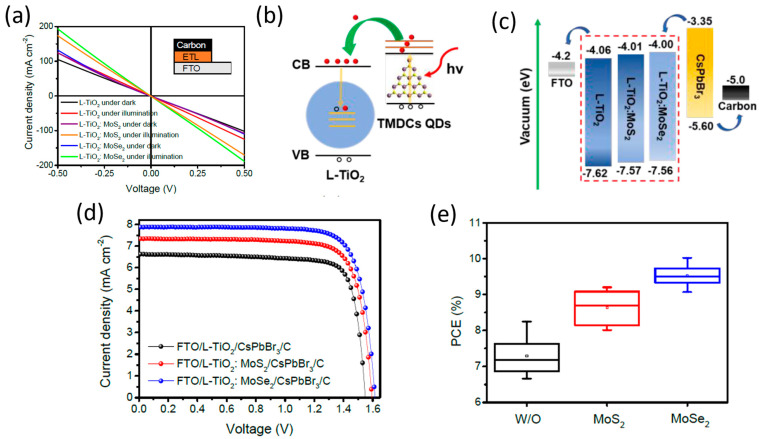
(**a**) J–V characteristics of the device FTO/L-TiO_2_:TMDCs QDs/C under dark and illumination (AM 1.5G); (**b**) Schematic diagram of the electron transfer from TMDCs QDs to TiO_2_ under illumination; (**c**) band alignment diagram of an all-inorganic CsPbBr_3_ PSC based on L-TiO_2_:TMDCs QDs ETLs; (**d**) J–V curves of various PSCs with and without TMDCs QDs; (**e**) statistical distribution of PCE for FTO/L-TiO_2_/CsPbBr_3_/C (black box), FTO/L-TiO_2_:MoS_2_/CsPbBr_3_/C (red box), and FTO/L-TiO_2_:MoSe_2_/CsPbBr_3_/C solar cells (blue box) [93].

**Figure 19 nanomaterials-14-01783-f019:**
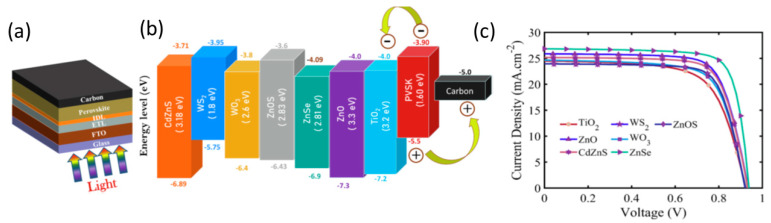
(**a**) Planar device structure; (**b**) Energy band diagram of FTO/ETM/Perovskite/C structure; (**c**) J–V curves for different ETMs [108].

**Figure 20 nanomaterials-14-01783-f020:**
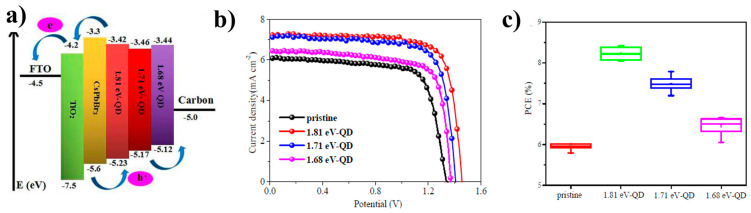
(**a**) Energy diagram of each layer in the device with energy levels given in eV; (**b**) J–V curves of inorganic PSCs with and without CuInS_2_/ZnS QDs under air mass 1.5 global (AM 1.5G, 100 mW cm^−2^) illumination; (**c**) Photovoltaic characteristics for 20 random PSC devices [110].

**Figure 21 nanomaterials-14-01783-f021:**
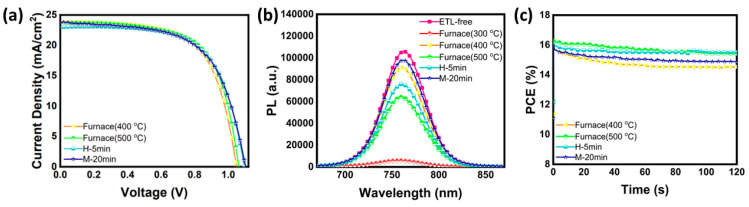
(**a**) Current density–voltage curves (in the reverse direction) related to glass/FTO/TiO_2_/perovskite/CuInS_2_/C structures; (**b**) Steady-state PL spectra of glass/mesoporous-TiO_2_/perovskite samples with mesoporous TiO_2_ layers cured under the H-lamp and M-lamp for optimized times of 5 and 20 min, respectively, in comparison with those of the samples that are cured in the furnace at temperatures of 300, 400, and 500 °C or do not have any ETL (ETL-free); (**c**) Steady-state PCE over 120 s for C-PSCs with both of their compact and mesoporous TiO_2_ layers cured using the H-lamp (H-5 min) and M-lamp (M-20 min) in comparison with samples with their compact and mesoporous TiO_2_ layers prepared in the furnace in temperatures of 400 and 500 °C [117] (reprinted with permission from ACS Appl. Energy Mater. 2021, 4, 8, 7800–7810. Copyright 2021 American Chemical Society).

**Figure 22 nanomaterials-14-01783-f022:**
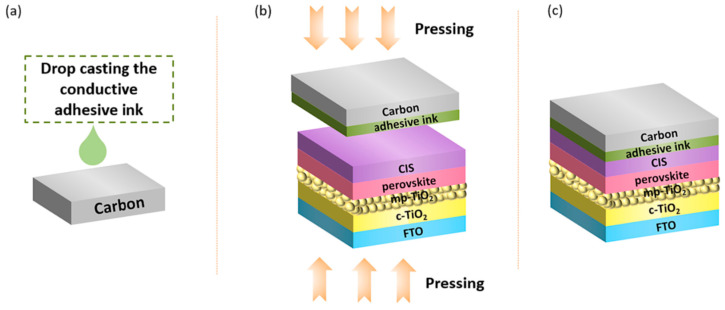
(**a**–**c**) Schematic illustration of direct lamination of carbon foil counter electrode on CIS HTL by conductive adhesive ink [119].

**Figure 23 nanomaterials-14-01783-f023:**
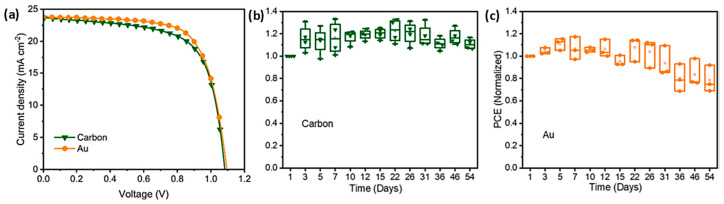
(**a**) J–V curves and (**b**,**c**) ambient long-term stability (25 °C; RH of 10 %) of PSCs based on laminated carbon with optimized ink layer and PSCs with Au electrode, respectively [119].

**Figure 24 nanomaterials-14-01783-f024:**
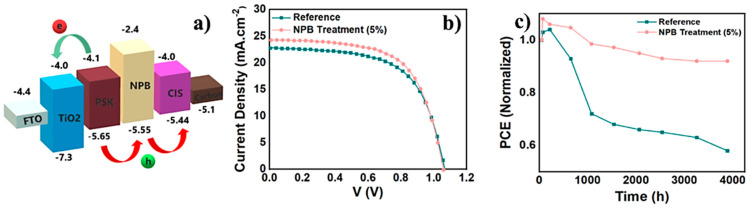
(**a**) Band diagram of mixed-perovskite devices with NPB post-treatment; (**b**) Current density–voltage (J–V) characteristic of devices; (**c**) Stability of respective photovoltaic devices out of glovebox and in ambient (RH = 45%) tracked by measurement of PCE for 4000 h [122].

**Figure 25 nanomaterials-14-01783-f025:**
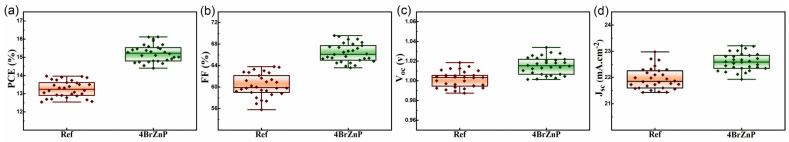
Statistical distribution of photovoltaic parameters of reference and 4BrZnP devices: (**a**) PCE; (**b**) FF; (**c**) Voc; (**d**) Jsc [123].

**Figure 26 nanomaterials-14-01783-f026:**
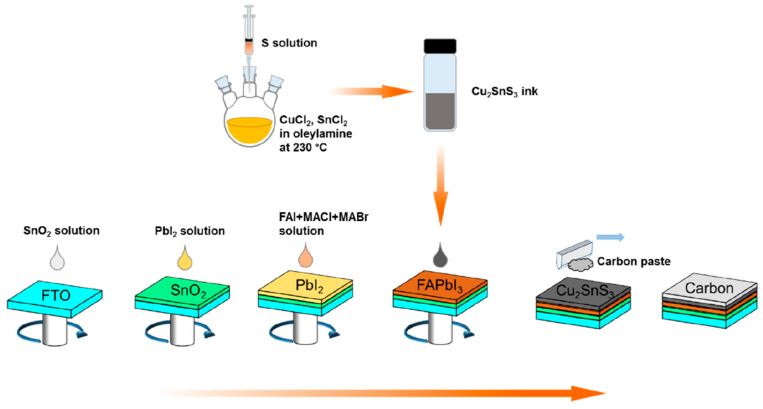
Schematic diagram of the preparation of Cu_2_SnS_3_ Ink and the fabrication process for the corresponding C-PSCs [124] (reprinted with permission from ACS Appl. Nano Mater. 2022, 5, 8, 10755–10762. Copyright 2022 American Chemical Society).

**Figure 27 nanomaterials-14-01783-f027:**
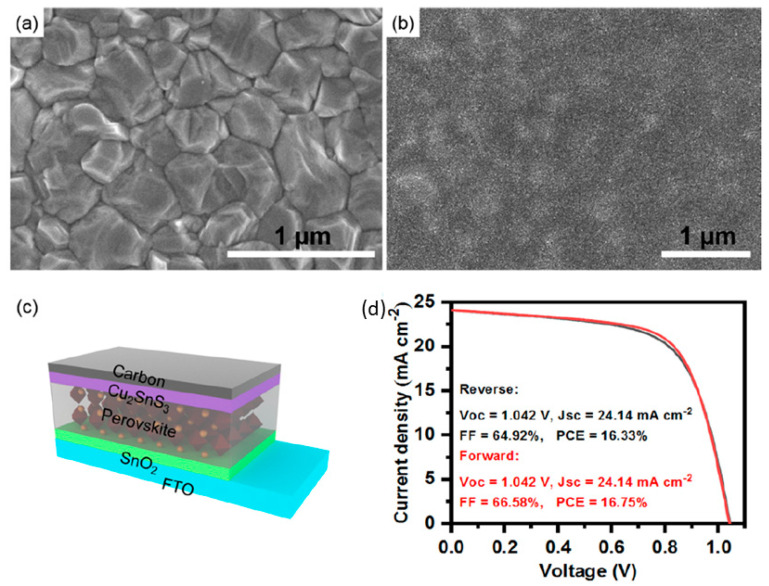
Planar-view SEM images of (**a**) FAPbI_3_ perovskite film and (**b**) Cu_2_SnS_3_ nanocrystal HTL; (**c**) Schematic device structures of a PSC; (**d**) J–V curves for PSCs with champion PCEs [124] (reprinted with permission from ACS Appl. Nano Mater. 2022, 5, 8, 10755–10762. Copyright 2022 American Chemical Society).

**Figure 28 nanomaterials-14-01783-f028:**
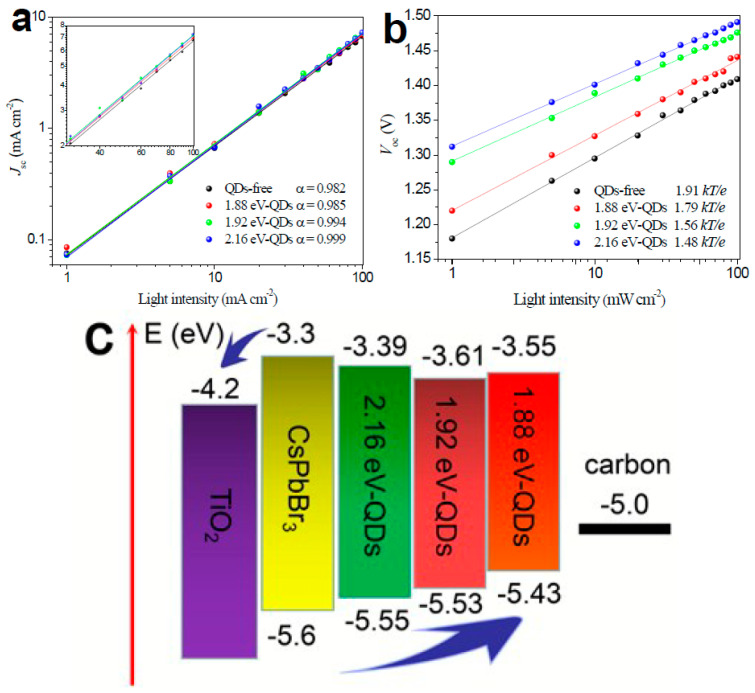
(**a**) Jsc and (**b**) Voc as a function of illuminated light intensity for the PSCs; (**c**) Energy diagram of each layer in the device with energy levels given in eV [125].

**Figure 29 nanomaterials-14-01783-f029:**
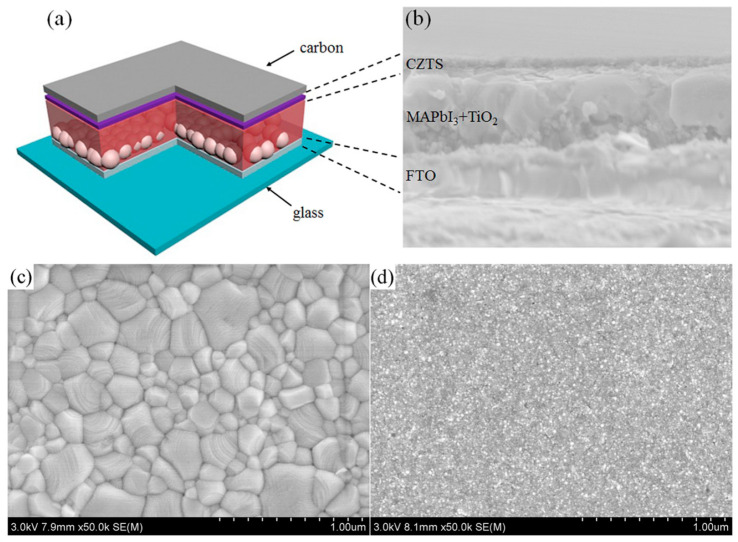
(**a**) Schematic device structure of the paintable carbon electrode-based perovskite solar cell; (**b**) Cross-sectional SEM image of a typical solar cell without a carbon electrode; (**c**) Planar SEM image of the perovskite film; (**d**) Planar SEM image of the CZTS film [129].

**Figure 30 nanomaterials-14-01783-f030:**
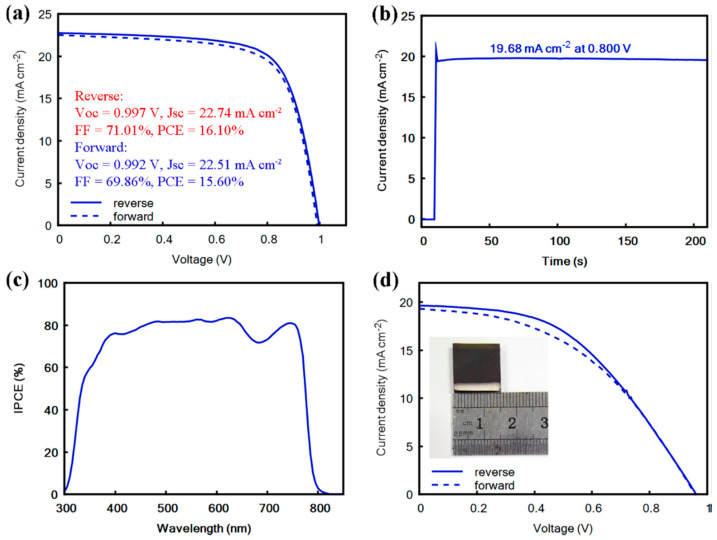
(**a**) Champion J–V curves; (**b**) Steady-state output current densities at maximum power point; (**c**) IPCE spectrum for perovskite solar cells employing gs-Cu_2_ZnSnS_4_ nanocrystals; (**d**) Typical J–V curves of a C-PSC employing gs-Cu_2_ZnSnS_4_ hole conducting layer with an effective area of 1.00 cm^2^. The inset shows the photograph of a 1.00 cm^2^ perovskite solar cell [131].

**Figure 31 nanomaterials-14-01783-f031:**
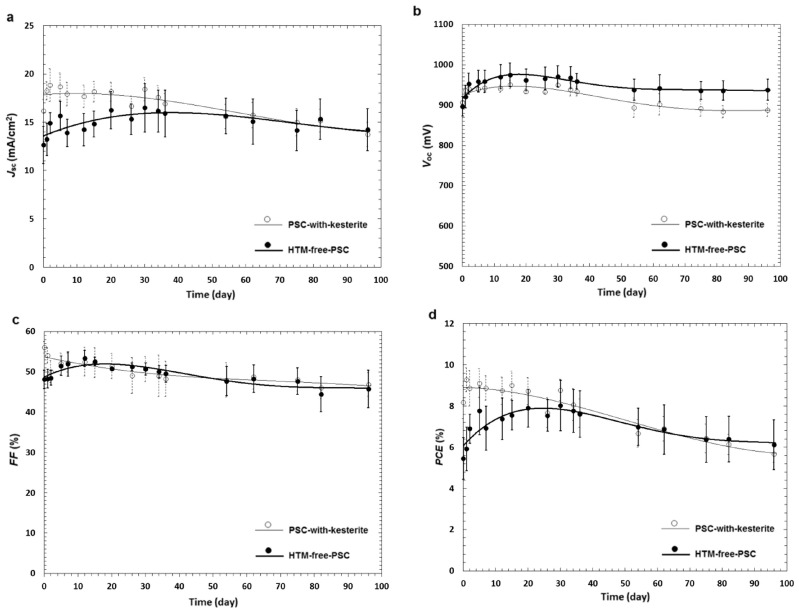
Stability diagrams of HTM-free-PSC and PSC-with-kesterite. The variation of (**a**) Jsc, (**b**) Voc, (**c**) FF, and (**d**) PCE with storing time. The samples were stored at room temperature, in the dark, and without encapsulation [132].

**Figure 32 nanomaterials-14-01783-f032:**
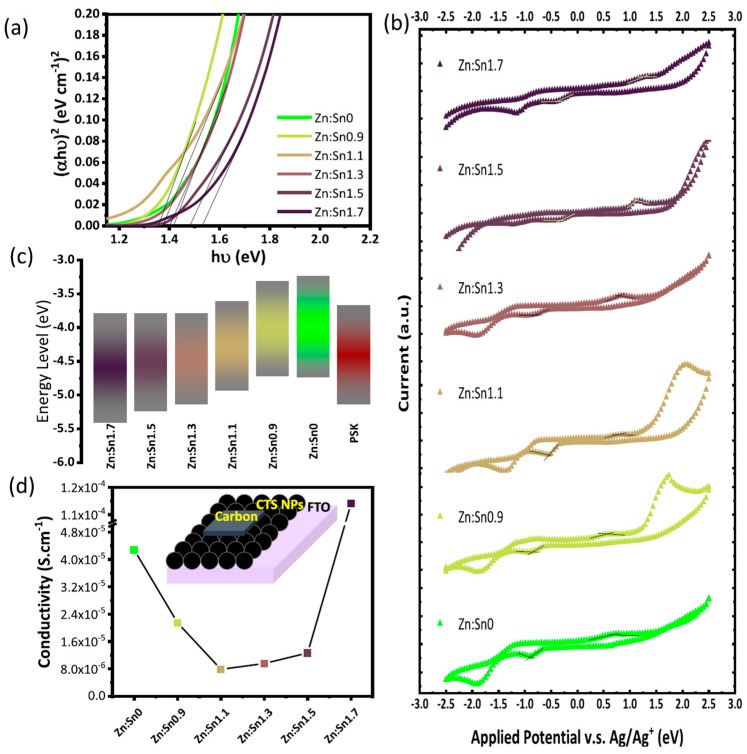
(**a**) Tauc plots of CZTS NPs measured from the transmittance spectra; (**b**) Cyclic voltammetry plots to estimate energy levels of CZTS NPs using 0.1 M TBAPF_6_ in acetonitrile as an electrolyte; (**c**) Schematic of the energy-level alignment between the PSK layer and different HTMs; (**d**) Electrical conductivity of CZTS thin films with different Zn/Sn ratios [133] (reprinted with permission from ACS Appl. Mater. Interfaces 2022, 14, 15, 17296–17311. Copyright 2022 American Chemical Society).

**Figure 33 nanomaterials-14-01783-f033:**
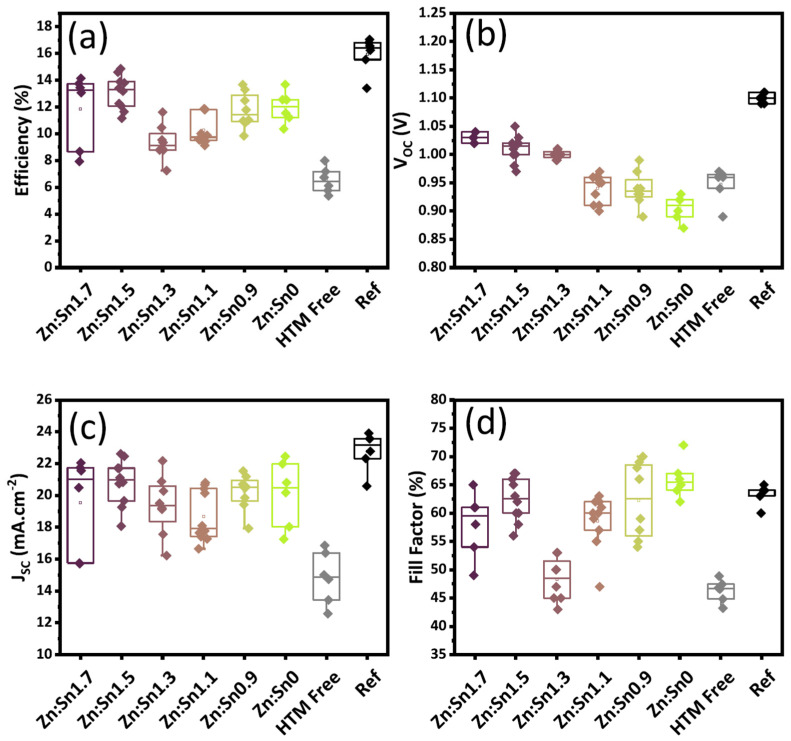
(**a**–**d**) Statistical box charts of photovoltaic parameters of PSCs based on CZTS HTMs with different ratios of Zn/Sn and CIS as the reference (efficiency, open-circuit voltage, short-circuit current density, and fill factor) [133] (reprinted with permission from ACS Appl. Mater. Interfaces 2022, 14, 15, 17296–17311. Copyright 2022 American Chemical Society).

**Figure 34 nanomaterials-14-01783-f034:**
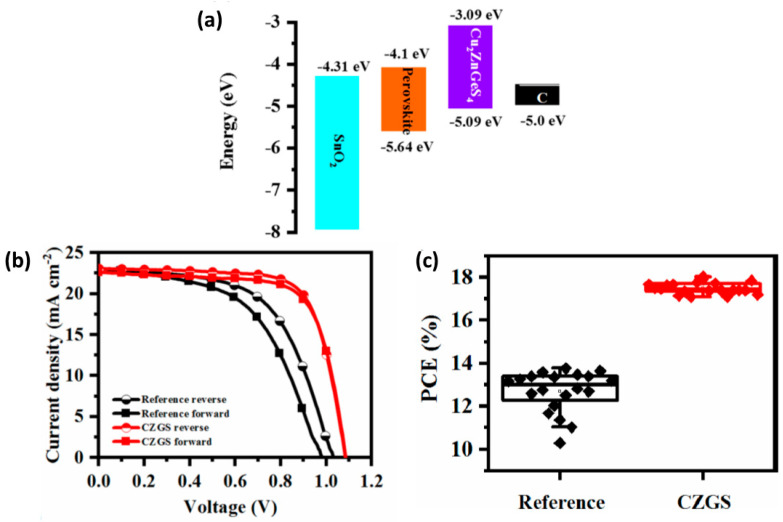
(**a**) Schematic energy-level diagram of the C-PSC; (**b**) J–V curves for the champion C-PSC with CZGS HTM and the reference HTM-free C-PSC; and (**c**) Summary of the PCE for the fabricated C-PSCs [135].

**Table 1 nanomaterials-14-01783-t001:** Summary of the most widely used chalcogenides and their optical and electrical properties.

Material	Energy Band Gap (eV)	Band Type	Carrier Mobility (cm^2^ V^−1^ s^−1^)	Reference
CuI	3.1	Direct	43.9 (h+)	[27]
Cu_x_S	1.6–2.2	Direct	1.17	[28]
MoS_2_	0.88–1.71 (bulk)	Indirect	30–500 (h+)	[29]
	1.72 (monolayer)	Direct	10–130 (e−)	
MoSe_2_	1.1 (bulk)	Indirect	90 (h+)	[30]
	1.5 (monolayer)	Direct	25 (e−)	
WS_2_	1.29 (bulk)	Indirect	50(h+)	[31]
	2.2 (monolayer)	Direct	200 (e−)	
WSe_2_	1.2 (bulk)	Indirect	250 (h+)	[29,32]
	1.7 (monolayer)	Direct	142 (e−)	
CuInS_2_	1.32–1.43	Direct	10.09	[33]
CuIn_1−x_Ga_x_S_2_	1.49–1.54	Direct	1.6–30.9	[34,35,36]
CuInxGa_(1−x)_Se_2_	1–1.7	Direct	3–22	[37]
BaZrS_3_	1.7	Direct	35	[38]
BaTiS_3_	1.3	Direct	25	[39]
CaZrS_3_	1.3	Direct	32	[40]
BaZrSe_3_	1.7	Direct	20	[41,42]

**Table 2 nanomaterials-14-01783-t002:** Summary of some of the highest-performing organic photovoltaics (OPVs) and perovskite solar cells (PSCs), which employ chalcogenides as hole transport materials (HTMs).

Chalcogenide	Solar Cell Type	Solar Cell Structure	PCE(%)	Reference
CuS	OPV	ITO/CuS/PTB7:PC_71_BM/Al	4.32	[59]
CuI	OPV	ITO/CuI/PBDTTPD:PC_61_BM/Sm/Al	5.54	[58]
WSe_2_	OPV	ITO/PEDOT:PSS/WSe_2_/PTB7:PC_71_BM/Al	8.5	[61]
In_2_Se_3_	OPV	ITO/In_2_Se_3_/PBDB-T:ITIC/Ca/Al	9.58	[53]
MoS_2_	OPV	ITO/MoS_2_/PBDB-T-2F:Y6:PC_71_BM/PFN-Br/Al	14.9	[55]
WS_2_	OPV	ITO/WS_2_/PBDB-T-2F:Y6/PFN-Br/Al	15.8	[56]
WS_2_	OPV	ITO/WS_2_/PBDB-T-2F:Y6:PC_71_BM/PFN-Br/Al	17	[56]
WS_2_	OPV	ITO/WS_2_/PBDB-T-2F:Y6:SF(BR)_4_/PFN-Br/Al	20.87	[57]
CuS	PSC	ITO/CuS/MAPbI_3_/C_60_/BCP/Ag	16.2	[62]
CuS	PSC	FTO/SnO_2_/MAPbI_3_/spiro-OMeTAD/Cu_x_S/Au	18.58	[63]
CuI	PSC	FTO/Na-TiO_2_/MAPbI_3_/CuI/Au	17.6	[64]
CuI	PSC	FTO/Cu@CuI/(CsFAMA)Pb(BrI)_3_/PC_61_BM/ZnO/Ag	18.8	[65]
CuI	PSC	ITO/Cu(Tu)I/MAPbI_3−x_Cl_x_/C_60_/BCP/Ag	19.9	[66]
WS_2_	PSC	ITO/WS_2_/MAPbI_3_/PCBM/BCP/Al	15	[67]
WS_2_	PSC	ITO/PTAA/WS_2_/(Rb_0.05_Cs_0.05_ FA_0.9_PbI_3_)_0.85_(MAPbBr_3_)_0.15_/C_60_/ZnSe/Cu(Ag)	20.92	[68]
MoS_2_	PSC	ITO/MoS_2_/MAPbI_3_/TiO_2_/Ag	20.43	[60]
PbS	PSC	FTO/TiO_2_/MAPbI_3_/PbS/Spiro-OMeTAD/Au	19.24	[69]
MnS	PSC	FTO/TiO_2_/MAPbI_3_/MnS/Au	19.86	[70]
CuZnS	PSC	ITO/P3CT-K/CuZnS/MAPbI_3_/PCBM/ZnO/Al	18.3	[71]
Cu_2_ZnSnS_4_	PSC	ITO/Cu_2_ZnSnS_4_/MAPbI_3_/PCBM/PrCMA/Ag	15.4	[72]
Cu_2_ZnSnSe_4_	PSC	FTO/Mo/Cu_2_ZnSnSe_4_/MAPbI_3_/ZnS/IZO/Ag	17.4	[73]

**Table 3 nanomaterials-14-01783-t003:** Summary table of the reported C-PSCs employing CuInS_2_ as the hole transport layer, which are presented in this work.

Power Conversion Efficiency (PCE) %	Device Configuration	Perovskite	Novelty–Highlights	Reference
18.5	FTO/SnO_2_/perovskite/CIS/C	Cs_0.05_(MA_0.17_FA_0.83_)_0.95_ Pb(I_0.83_ Br_0.17_)_3_	4BrZnP porphyrin additive in the perovskite	[123]
17.2	FTO/c-TiO_2_/m-TiO_2_/perovskite/CIS/conductive adhesive ink/C	Cs_0.05_(MA_0.17_FA_0.83_)_0.95_Pb(I_0.83_Br_0.17_)_3_	⮞ Carbon lamination with conductive adhesive ink comprising PMMA, CIS, and carbon black ⮞ 92% PCE stability over 54 days’ shelf-life	[119]
16.5	FTO/SnO_2_/perovskite/CIS/C	Cs_0.05_(MA_0.17_FA_0.83_)_0.95_Pb(I_0.83_Br_0.17_)_3_	Treatment of SnO_2_ ETL with urea	[114]
16.3	FTO/c-TiO_2_/m-TiO_2_/perovskite/CIS/C	Cs_0.05_(MA_0.17_FA_0.83_)_0.95_Pb(I_0.83_Br_0.17_)_3_	⮞ Replacement of sintering process, with light-curing procedure for TiO2 bilayer ⮞ Printable CuInS2/C hole collector	[117]
16.11	FTO/TiO_2_/perovskite/NPB/CIS/C	Cs_0.05_ (MA_0.16_FA_0.79_)_0.95_ Pb (I_0.84_Br_0.16_)_3_	⮞ N,N′-di(naphthalene-1-yl)-N,N′-diphenyl-benzidine (NPB) passivation layer ⮞ 92% PCE stability at 4000 h shelf life	[122]
16	FTO/c-TiO_2_/m-TiO_2_/perovskite/CIS/C	Cs_0.05_(MA_0.17_FA_0.83_)_0.95_Pb(I_0.83_Br_0.17_)	⮞ Polytriarylamine (PTAA) doping of CIS ⮞ 100% PCE stability at 1720 h shelf life ⮞ 70% PCE stability after 408 h in a thermally stressed ambient medium (60 °C-40% humidity)	[120]
13.14	FTO/c-TiO_2_/m-TiO_2_/perovskite/CIS/C	CH_3_NH_3_PbI_3_	Multilayered SnO_2_-TiO_2_ ETL	[115]
13.09	FTO/c-TiO_2_/m-TiO_2_/perovskite/CIS/C	CH_3_NH_3_PbI_3_	⮞ TiO_2_ thickness optimization ⮞ Fully ambient-processed	[116]
11.24	FTO/c-TiO_2_/m-TiO_2_/perovskite/PTSAx-y/CIS/C	MAPbI_3_	*p*-toluene sulfonamide (PTSA) passivation layer	[121]
10.85	FTO/c-TiO_2_/m-TiO_2_/CIS-Zn QDs/C	CsPbBr_3_	⮞ Core/shell structured CuInS2/ZnS QDs ⮞ Incorporation of a long persistence phosphor (LPP) into the carbon ink for reabsorption-conversion of low-energy photons	[111]
10.16	FTO/SnO_2_/perovskite/CIS/C	Cs_0.05_(MA_0.17_FA_0.83_)_0.95_Pb(I_0.83_Br_0.17_)_3_	Electrochemical deposition of perovskite	[113]
9.93	FTO/c-TiO_2_/m-TiO_2_/perovskite/CIS/C	Cs_0.05_(MA_0.17_FA_0.83_)_0.95_Pb(I_0.83_Br_0.17_)_3_	⮞ Slot-die coated CIS HTL ⮞ Fully printed PSCs	[118]
9.43	FTO/c-TiO_2_/m-TiO_2_/GQDs/perovskite/CIS-ZnS/C	CsPbBr_3_	⮞ Employment of graphene QDS and CuInS2/ZnS ⮞ Bifunctional solar energy and water vapor energy-harvesting device ⮞ 98% PCE stability after 40 days’ shelf-life	[112]
8.42	FTO/cTiO_2_/m-TiO_2_/CIS-ZnS/C	CsPbBr_3_	⮞ CuInS_2_/ZnS quantum dots on the film of perovskite ⮞ 94% PCE stability in high-humidity conditions	[110]

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
