# Peer review of "Chalcogenides in Perovskite Solar Cells with a Carbon Electrode: State of the Art and Future Prospects"

_nanomaterials, 2024, doi:10.3390/nano14221783_

Round 1
Reviewer 1 Report
Comments and Suggestions for Authors
The review provides a comprehensive overview of the incorporation of chalcogenides in perovskite solar cells (PSCs) with a focus on carbon-based electrodes. The structure is well-defined, beginning with an introduction to PSCs, the use of chalcogenides in various roles, and culminating with prospects and challenges. It is an informative piece for those researching advanced materials for photovoltaic applications, and it highlights the most recent advancements in this field. However, the content needs to be further supplemented and improved. I recommend accepting this manuscript after the following issues are properly addressed.
1. The introduction lacks a distinction between this review and related reviews that have already been published.
2. The introduction of carbon electrodes in the field of perovskite in the second part of the article is too limited. It is suggested to supplement and enrich it.
3. Some figures in the article are unclear and need to be adjusted. Some of the images in the article are labeled as “(a)” or “a” while others are labeled as “a)”, and the font sizes are also different. It is recommended to maintain consistency.
4. Check the article format, for example, on line 441, “CsPbBr3” should be “CsPbBr3”.
5. At the end of each section, there is a lack of summary, and the overview should reflect more of the author's ideas.
6. It is suggested to add specific research questions or challenges that need to be addressed for future research directions in the conclusion section, which will make the section more inspiring to readers.
7. Some references need to be cited, which has a certain promoting effect on improving the quality of the article (DOI: 10.1021/acscatal.0c00847, DOI: 10.1002/anie.202114450, DOI: 10.1002/anie.202215136, DOI: 10.1021/jacs.2c03875).
8. Some paragraphs have a first line indentation, while others do not. The format and details of the entire text need to be improved.
Author Response
The review provides a comprehensive overview of the incorporation of chalcogenides in perovskite solar cells (PSCs) with a focus on carbon-based electrodes. The structure is well-defined, beginning with an introduction to PSCs, the use of chalcogenides in various roles, and culminating with prospects and challenges. It is an informative piece for those researching advanced materials for photovoltaic applications, and it highlights the most recent advancements in this field. However, the content needs to be further supplemented and improved. I recommend accepting this manuscript after the following issues are properly addressed.
- The introduction lacks a distinction between this review and related reviews that have already been published.
Reply: In the Introduction part (lines 69-74) the purpose and motivation of this work are mentioned, that dissociate this review from the related reviews in literature.
“This review intends to present the latest advancements of the incorporation of chalcogenides in PSCs with a C electrode. The motivation has been on one hand the highly promising results that have been obtained by the implementation of chalcogenides in PSCs with a metal electrode, that achieve high performance and stability, and on the other hand the noticed increasing number of reports and increasing funding on the research on chalcogenides in PSCs, which is currently moving to C-PSCs as well.”
The following additional explanation has been added to the text, as proposed
“The focus on PSCs with a C electrode, a PSC structure that is gaining increasing amount of attention, owing to the favorable properties which makes it the most suitable and viable choice for the future commercialization of large area PSCs, is what distinguishes this review from the related reviews that already exist in literature. The lack of any relevant report so far is a gap in literature that is intended to be bridged by this work.”
- The introduction of carbon electrodes in the field of perovskite in the second part of the article is too limited. It is suggested to supplement and enrich it.
Reply: The part of carbon electrodes in perovskite solar cells has been enriched with the following text and related references, to which the readers can refer to for further study of the subject
“The two main configurations of C-PSCs are distinguished by the annealing temperature that is applied for the preparation of the C electrode, and they include the low temperature C-PSCs (LT-CPSCs) and high temperature C-PSCs (HT-CPSCs) respectively. In LT-CPSCs a C paste is deposited on the HTL by painting, blade coating or printing methods and the electrode is formed after an-nealing at temperatures below 100 degrees. On the other hand, in HT-CPSCs, a triple mesoscopic stack is formed by a mesoporous ETL, typically TiO2, a meso-porous insulating layer, that prevents the contact of the resulting counter electrode with the ETL, typically ZrO2, and a porous, thick C electrode that is deposited on top of the TiO2/ZrO2 bilayer, mostly by blade coating, in the form of C paste which is further treated at high temperatures (350-400 degrees). The perovskite is then inserted in C-PSCs by infiltration of a small quantity of perovskite precursor solu-tion through the triple mesoscopic stack and the perovskite crystals are formed with subsequent annealing. Despite the major differences in the device configura-tion, the two structures have some common features, which are of great importance for the transfer of this technology to the large area and the commercialization of PSCs of this type: the deposition methods are simple, low-cost and industrially compatible, while they also achieve a minimum waste of materials, therefore a low cost of fabrication. Both configurations have also proven to be efficient after the elimination of the HTL, which has a great effect on both the cost and the stability of the devices, while they can be prepared entirely under ambient conditions, with no control facilities (e.g. glove box, clean room) required. Moreover, the chemical in-ertness of carbon, which prevents the degradation of the electrode overtime, com-bined with the hydrophobic nature of carbon, which prevents the degradation of the perovskite layer, give a significant advantage of C-PSCs over metal electrode PSCs, since they have proven to be of high thermal, environmental and operational stability[14–17]. Adding to the above, the considerably lower cost of C electrodes compared to metal electrodes, C-PSCs are the most suitable candidates for com-mercially available perovskite solar modules (PSMs), which is also proved by the increasing number of PSM fabrication companies that switch to C electrodes in order to achieve a viable product.”
- Some figures in the article are unclear and need to be adjusted. Some of the images in the article are labeled as “(a)” or “a” while others are labeled as “a)”, and the font sizes are also different. It is recommended to maintain consistency.
Reply: We thank the reviewer for the thorough screening of our manuscript.
The captions of Figures 10,22 and 34 have been corrected and Figure 34 has been modified for clarity.
- Check the article format, for example, on line 441, “CsPbBr3” should be “CsPbBr3”.
Reply: The manuscript has been revised and double checked as per any sub/superscript omission or mistake.
- At the end of each section, there is a lack of summary, and the overview should reflect more of the author's ideas.
Reply: A summary has been added to the end of each Section (4.1 Sulfides, 4.2 Selenides, 4.3 Ternary and 4.4 Quaternary), as suggested by the reviewer. The overview has also been enriched with more of the authors’ ideas.
- It is suggested to add specific research questions or challenges that need to be addressed for future research directions in the conclusion section, which will make the section more inspiring to readers.
Reply: A paragraph has been added to the Conclusion section, which combined with the enrichment of the Prospects and Challenges section, we believe that will meet the reviewer’s requirements.
- Some references need to be cited, which has a certain promoting effect on improving the quality of the article (DOI: 10.1021/acscatal.0c00847, DOI: 10.1002/anie.202114450, DOI: 10.1002/anie.202215136, DOI: 10.1021/jacs.2c03875).
Reply: The proposed references have been added in the manuscript, in the mention of chalcogenides as catalysts.
- Some paragraphs have a first line indentation, while others do not. The format and details of the entire text need to be improved.
Reply: Spelling corrections have been made, the format, spacing, paragraph style and alignment of Figure captions and paragraphs have been reformed to be homogeneous throughout the text.
Reviewer 2 Report
Comments and Suggestions for Authors
While this paper may be considered for further review, I believe that the following questions are important to address transparently. I say so because I cannot follow whether the following points are adequately placed so the readers can follow?
In the context of PSCs utilizing carbon electrodes, how do the stability and power conversion efficiencies (PCEs) compare with traditional metal back contacts, and what specific mechanisms contribute to any observed differences in degradation rates under environmental stressors?
How do the electronic and structural properties of chalcogenides influence charge transport and recombination processes within PSCs, and what specific characteristics make certain chalcogenides more effective than others when incorporated into perovskite layers or interlayers? What kind of non-covalent interactions are involved?
What challenges arise in optimizing the interfaces between chalcogenide materials and perovskite layers in PSCs with carbon electrodes, and what experimental strategies can be employed to mitigate issues related to energy level alignment and interfacial stability?
Considering the move toward commercializing C-PSCs, what are the main technical and economic barriers to scaling up the production of chalcogenide-infused PSCs, and how might these barriers differ from those faced by traditional PSCs?
In the implementation of chalcogenides in PSCs, what are the key synergistic effects observed when combining these materials with carbon electrodes, and how can these interactions be quantitatively measured to predict overall device performance improvements?
Background references should be updated, together with the improvement of the writing quality of the ms.
Comments on the Quality of English Language--
Author Response
While this paper may be considered for further review, I believe that the following questions are important to address transparently. I say so because I cannot follow whether the following points are adequately placed so the readers can follow?
- In the context of PSCs utilizing carbon electrodes, how do the stability and power conversion efficiencies (PCEs) compare with traditional metal back contacts, and what specific mechanisms contribute to any observed differences in degradation rates under environmental stressors?
Reply: We thank the reviewer for the consideration of our manuscript. An introduction to the advantages of C-PSCs over the typical metal contact PSCs and the reasons for their superiority has been presented in both the Introduction Part and Part 2 of the manuscript, where a mention to the PCEs obtained in C-PSCs has been made, followed by a Figure showing the PCE evolution over the years. We believe that, in an already length article, where the scope is the presentation of chalcogenides, further analysis of the mechanisms behind the increased stability of C electrodes over the metal electrodes is out of the purpose of our work.
However, in order to improve the second part of the manuscript, where the advantages of C electrodes are mentioned, the following text along with the relevant references, to which the readers can refer to for further study of the subject, has been added
“The two main configurations of C-PSCs are distinguished by the annealing temperature that is applied for the preparation of the C electrode, and they include the low temperature C-PSCs (LT-CPSCs) and high temperature C-PSCs (HT-CPSCs) respectively. In LT-CPSCs a C paste is deposited on the HTL by painting, blade coating or printing methods and the electrode is formed after annealing at temperatures below 100 degrees. On the other hand, in HT-CPSCs, a triple mesoscopic stack is formed by a mesoporous ETL, typically TiO2, a meso-porous insulating layer, that prevents the contact of the resulting counter electrode with the ETL, typically ZrO2, and a porous, thick C electrode that is deposited on top of the TiO2/ZrO2 bilayer, mostly by blade coating, in the form of C paste which is further treated at high temperatures (350-400 degrees). The perovskite is then inserted in C-PSCs by infiltration of a small quantity of perovskite precursor solu-tion through the triple mesoscopic stack and the perovskite crystals are formed with subsequent annealing. Despite the major differences in the device configuration, the two structures have some common features, which are of great importance for the transfer of this technology to the large area and the commercialization of PSCs of this type: the deposition methods are simple, low-cost and industrially compatible, while they also achieve a minimum waste of materials, therefore a low cost of fabrication. Both configurations have also proven to be efficient after the elimination of the HTL, which has a great effect on both the cost and the stability of the devices, while they can be prepared entirely under ambient conditions, with no control facilities (e.g. glove box, clean room) required. Moreover, the chemical inertness of carbon, which prevents the degradation of the electrode overtime, combined with the hydrophobic nature of carbon, which prevents the degradation of the perovskite layer, give a significant advantage of C-PSCs over metal electrode PSCs, since they have proven to be of high thermal, environmental and operational stability [14–17]. Adding to the above, the considerably lower cost of C electrodes compared to metal electrodes, C-PSCs are the most suitable candidates for commercially available perovskite solar modules (PSMs), which is also proved by the increasing number of PSM fabrication companies that switch to C electrodes in order to achieve a viable product.”
- How do the electronic and structural properties of chalcogenides influence charge transport and recombination processes within PSCs, and what specific characteristics make certain chalcogenides more effective than others when incorporated into perovskite layers or interlayers? What kind of non-covalent interactions are involved?
Reply: We thank the reviewer for posing some very interesting questions. The detailed response for these questions can be found for each different chalcogenide in the Results and Discussion part of the corresponding references, where experimental data have proven the beneficial effect of chalcogenides, which has been explained in each individual case. The thorough explanation of the mechanisms that contribute to these effects are out of the scope of this review, where the main objective is to present the latest advancements in literature, in order to intrigue the readers for a further reading and research on this topic.
3.What challenges arise in optimizing the interfaces between chalcogenide materials and perovskite layers in PSCs with carbon electrodes, and what experimental strategies can be employed to mitigate issues related to energy level alignment and interfacial stability?
Reply: The type of measures that need to be taken and the experimental strategies that can be employed in order to optimize both the energy level alignment and the obtaining of an interface with the desirable characteristics, that would reduce the recombination of charges and the series resistance of C-PSCs, which are the main obstacles that need to be tackled to improve the performance of these devices, and in particular the Voc and FF values, which remain quite lower than the respective values of metal electrode PSCs is highly dependent on several factors. To name a few: 1. the device structure (e.g. HT-CPSCs or LT-CPSCs, planar or mesoporous, HTL-free or HTL-based, ambient processed or not etc.), 2. the individual layer components and materials used (e.g. type of ETL, type of perovskite, type of HTL if used etc.), 3. the fabrication method used (e.g. spin coating, printing, evaporation, blade coating etc.) and many more. The response to this question is too multifactorial to be addressed in a review article. To gain more knowledge and to derive the response to this matter, the readers can refer to the cited literature, which is plenty, in each case of interest.
4.Considering the move toward commercializing C-PSCs, what are the main technical and economic barriers to scaling up the production of chalcogenide-infused PSCs, and how might these barriers differ from those faced by traditional PSCs?
Reply: Section 5. Prospects and Challenges and Section 6. Conclusions have been enriched and we believe that with these additions the response to this question has been addressed. However, it should be pointed out that there has been no report yet on the attempt of chalcogenides implementation in large area C-PSCs, which would define the exact problems, both technical and financial, that need to be resolved in order for these materials to meet the requirements for low-cost, large area C-PSCs. As in all materials in their starting studies, a lot of effort needs to be put towards the application of chalcogenides in solar cells, using industrially compatible methods, before reaching to accurate results.
With respect to the differences with traditional PSCs, considering the difference in the fabrication methods of C-PSCs versus metal electrode PSCs, which have been reported in the manuscript, there can be no direct comparison, and if there were, it would be out of the scope of this manuscript and suitable for a different manuscript, where the scope would be the comparison of typical PSCs and C-PSCs from a material point of view. In our work we intend to present a summary of chalcogenide application in C-PSCs, rather that proceeding to any comparison with metal electrode PSCs.
5.In the implementation of chalcogenides in PSCs, what are the key synergistic effects observed when combining these materials with carbon electrodes, and how can these interactions be quantitatively measured to predict overall device performance improvements?
Reply: Initially, the basic point that needs to be taken into account when applying chalcogenides in C-PSCs is their interaction and energy level alignment with the other components of a C-PSC and mainly the perovskite in use. A second point to be carefully tailored is the implementation method, since C-PSCs have a different structure and different fabrication parameters than metal electrode PSCs. Therefore, the key parameter for a successful incorporation regards the modification of the C/Chalcogenide interface fabrication method, to achieve uniformity and a complete coupling of materials, with no surface defects that could act as recombination sites. The C electrode itself is rather inert, and with a work function very close to the typical metal electrodes (work function values: C=4.8 eV, Au =5.1 eV and Ag =4.73 eV). However, the creation of C-chalcogen bonds has been reported (e.g. C-S and C-Se) which increase the conductive paths for charges and enhance the electrocatalytic activity of C electrodes. These studies have mainly been conducted for energy storage applications (for instance https://doi.org/10.1039/D2TA00269H, https://doi.org/10.1002/smll.201800148, https://doi.org/10.1002/aenm.201800927), but it is a good opportunity, with the rising interest in chalcogenide applications in PSCs, for the detailed mechanisms to be investigated in experimental studies, for perovskite solar cells with a C electrode as well. This is one of the scopes of this manuscript, to provide an insight for readers and drive their interest into experimentally investigating more in depth these highly promising materials.
- Background references should be updated, together with the improvement of the writing quality of the ms.
Reply: The manuscript has been revised and double checked as per any sub/superscript omission or mistake. Spelling corrections have been made, the format, spacing, paragraph style and alignment of Figure captions and paragraphs have been reformed to be homogeneous throughout the text.
Reviewer 3 Report
Comments and Suggestions for Authors
The authors review the use of calcogenides as electrode materials in perovskite solar cells containing a carbon electrode. The paper is on an interesting topic and the number of specific materials reviewed rather wide.
As only concern the authors should properly check the text in order to remove few typos .
Author Response
The authors review the use of chalcogenides as electrode materials in perovskite solar cells containing a carbon electrode. The paper is on an interesting topic and the number of specific materials reviewed rather wide.
As only concern the authors should properly check the text in order to remove few typos.
Reply: We thank the reviewer for the consideration of our work. In order to improve the quality of our manuscript, the manuscript has been revised and double checked as per any sub/superscript omission or mistake, spelling and typo corrections have been made and the format, spacing, paragraph style and alignment of Figure captions and paragraphs have been reformed to be homogeneous throughout the text.
Reviewer 4 Report
Comments and Suggestions for Authors
This manuscript describes chalcogenides in perovskite solar cells with a carbon electrode. The review article is lengthy and the content is enough. It is suggested to publish this review article after a major revision. Some comments are described as following.
1. Some figures are not clear, please check. For example, Figure 3 and figure 6.
2. The authors have exhibit lots of chalcogenides, which are used in the perovskite solar cells. Please discuss the performance trend in among these chalcogenides, rather than list the devices and performance parameters.
3. Indeed, there are many works include chalcogenides in the perovskite solar cells. However, the perovskite solar cell can also be independent on the chalcogenides, therefore, the key influence of the chalcogenides should be discussed in the term of the microstructure and electronic structures. Please enrich this part.
4. Some typo error should be checked, such as PSCS, the arrow in the end of the title. Please check the whole content carefully to avoid these mistakes.
5. In conclusion part, the authors should point out the problems and challenges in current research progress. And further discuss the possible strategies to solve these problems. Please revise this part.
6. Some optoelectronic progress are suggested to be included in the introduction part, such as https://doi.org/10.1007/s12598-019-01261-y; https://doi.org/10.1007/s12598-021-01909-8.
Author Response
This manuscript describes chalcogenides in perovskite solar cells with a carbon electrode. The review article is lengthy and the content is enough. It is suggested to publish this review article after a major revision. Some comments are described as following.
- Some figures are not clear, please check. For example, Figure 3 and Figure 6.
Reply: Figure 3 has been replaced for clarity.
- The authors have exhibit lots of chalcogenides, which are used in the perovskite solar cells. Please discuss the performance trend in among these chalcogenides, rather than list the devices and performance parameters.
Reply: The text below has been added to Section 3 of the manuscript, to summarize the trend among the chalcogenides used in solar cells, serving as an introductory paragraph for the analysis that follows.
“A variation of the metallic element in the chalcogenide, typically combined with sulfur (S) and selenium (Se), results in variations in the optoelectronic properties of the resulting chalcogenide, with prominent effects on the energy levels and the type of mobility of the resulting materials. The performance of chalcogenides’ application in solar cells exhibits the following trend:
- Transition metal chalcogenides (TMDs), such as molybdenum disulfide (MoS2), molybdenum diselenide (MoSe2) and tungsten diselenide (WSe2), have a superior hole mobility while the can also produce homogeneous films with tunable properties, and have been, therefore, mainly used as hole transport materials (HTMs) to prepare the hole transport layers (HTLs) for organic photovoltaics (OPVs) and perovskite solar cells (PSCs).
- Metal chalcogenides, such as lead (Pb) based and copper (Cu) based ternary and quaternary, together with transition metal cadmium (Cd) based and the semi-metal antimony (Sb) based chalcogenides, have exhibited excellent performance as absorbers for thin film solar cells and quantum dot solar cells (QDSSCs).
- Transition metal cadmium (Cd) based sulfides and selenides have exhibited satisfactory performance as electron transport layers (ETLs), applied in perovskite solar cells (PSCs), organic photovolatics (OPVs) and antimony chalcogenide solar cells (Sb-CSCs).
- Transition metal cobalt (Co) and nickel (Ni) sulfides, along with metal copper (Cu) sulfide chalcogenides have achieved high performance photoelectrochemical solar cells, which includes dye sensitized solar cells (DSSCs) and quantum dot sensitized solar cells (QDSSCs), when applied as counter electrodes.”
- Indeed, there are many works include chalcogenides in the perovskite solar cells. However, the perovskite solar cell can also be independent on the chalcogenides, therefore, the key influence of the chalcogenides should be discussed in the term of the microstructure and electronic structures. Please enrich this part.
Reply: The following paragraph has been added to Section 3.2. Implementation in Solar Cells, in order to introduce the readers to the electronic and structural properties of chalcogenides that can have an effect on solar cell performance, and to refer them to some review papers, where a comprehensive analysis can be found. The detailed properties of chalcogenides that have an influence in optoelectronic devices, including solar cells, are too extensive and far from the scope of the present manuscript.
“Metal chalcogenides, including metal, transition metal, semi-metal and rare-earth metal chalcogenides, irrespectively of the number of constituents and chalcogens, possess electronic structures that can be tuned through a series of methods, some of which are thickness modification, defect engineering, intercalation of atoms, molecules, and ions in their lattice and chemical composition modifications. All of the aforementioned alterations have an impact in their optoelectronic properties, which consequently affect and determine their performance, once implemented in solar cells. More specifically, the quantum confinement effect that appears when moving from bulk chalcogenides to low dimensional chalcogenide structures (e.g. 2D) is responsible for i. the transition of their bandgap from indirect to direct, with a simultaneous enlargement and ii. the appearance of strong excitonic effects, attributed to the stronger electrostatic force interactions at lower thicknesses. On the other hand, defect engineering can lead to higher photoluminesence intensities, while intercalation of various species in the lattice of chalcogenides can induce the transition from semiconducting to conducting and enhance optical transmission. Finally, by alloying metallic and semiconducting chalcogenides and fine tuning their chemical composition, a fine tuning of the resulting alloy’s bandgap can be achieved.”
- Some typo error should be checked, such as PSCS, the arrow in the end of the title. Please check the whole content carefully to avoid these mistakes.
Reply: We thank the reviewer for the consideration of our work. In order to improve the quality of our manuscript, the manuscript has been revised and double checked as per any sub/superscript omission or mistake, spelling and typo corrections have been made and the format, spacing, paragraph style and alignment of Figure captions and paragraphs have been reformed to be homogeneous throughout the text.
- In conclusion part, the authors should point out the problems and challenges in current research progress. And further discuss the possible strategies to solve these problems. Please revise this part.
Reply: Section 5. Prospects and Challenges and Section 6. Conclusions have been enriched and we believe that with these additions the reviewer’s comment has been addressed.
- Some optoelectronic progress are suggested to be included in the introduction part, such as https://doi.org/10.1007/s12598-019-01261-y; https://doi.org/10.1007/s12598-021-01909-8; https://doi.org/10.1002/EXP.20230011; Journal of Inorganic Materials, 2023, 38(9): 1055-1061; https://doi.org/10.1002/adma.202400365.
Reply: We thank the reviewer for the suggestions that intend to strengthen our manuscript.
Reference https://doi.org/10.1002/EXP.20230011 has been included in the reference list in Line 207, where catalysis is referred to as an application of chalcogenides.
Reference https://doi.org/10.1007/s12598-021-01909-8 has been included in the reference list in Line 241, where the thickness dependence of optoelectronic properties of 2D materials is referred.
Reference https://doi.org/10.1002/adma.202400365 has been included in the reference list in Line 241, where the quantum confinement effect of 2D materials is referred, together with a brief mention on their favorable and tunable electronic structure.
Reference https://link.springer.com/article/10.1007/s12598-019-01261-y is dealing with a topic which is out of the scope of this manuscript.
Round 2
Reviewer 2 Report
Comments and Suggestions for Authors
Authors have revised their paper as suggested, and this wok may be suitable for publication
Comments on the Quality of English Language--
Reviewer 4 Report
Comments and Suggestions for Authors
The authors have answered all questions, thus an acceptance is suggested.